



# 1 Object-based ensemble estimation of snow depth and snow water
# 2 equivalent over multiple months in Sodankylä, Finland

David Brodylo[1], Lauren V. Bosche[1], Ryan R. Busby[2], Elias J. Deeb[3], Thomas A. Douglas[1], Juha
Lemmetyinen[4]
[1]U.S. Army Cold Regions Research and Engineering Laboratory, Fort Wainwright, AK 99709, USA
[2]U.S. Army Construction Engineering Research Laboratory, Champaign, IL 61826, USA
[3]U.S. Army Cold Regions Research and Engineering Laboratory, Hanover, NH 03755, USA
[4]Finnish Meteorological Institute, 00101 Helsinki, Finland
*Correspondence to*: David Brodylo (david.brodylo@usace.army.mil)
**Abstract.** Snowpack characteristics such as snow depth and snow water equivalent (SWE) are widely studied in regions prone
to heavy snowfall and long winters. These features are measured in the field via manual or automated observations and over
larger spatial scales with stand-alone remote sensing methods. However, individually these methods may struggle with
accurately assessing snow depth and SWE in local spatial scales of several square kilometers. One method for leveraging the
benefits of each individual dataset is to link field-based observations with high-resolution remote sensing imagery and then
employ machine learning techniques to estimate snow depth and SWE across a broader geographic region. Here, we combined
field-based repeat snow depth and SWE measurements over six instances from December 2022 to April 2023 in Sodankylä,
Finland with Light Detection and Ranging (LiDAR) and WorldView-2 (WV-2) data to estimate snow depth, SWE, and snow
density over a 10 km$^2$ local scale study area. This was achieved with an object-based machine learning ensemble approach by
first upscaling more numerous snow depth field data and then utilizing the estimated local scale snow depth to aid in estimating
SWE over the study area. Snow density was then calculated from snow depth and SWE estimates. Snow depth peaked in
March, SWE shortly after in early April, and snow density at the end of April. The ensemble-based approach had encouraging
success with upscaling snow depth and SWE. Associations were also identified with carbon- and mineral-based forest surface
soils, alongside dry and wet peatbogs.

## 24 1 Introduction

Seasonal snow is found in regions of the globe that experience freezing temperatures and is widely studied to monitor
changes in climate and hydrology. Snow is a component of the cryosphere that is heterogeneous over space and time. Snowmelt
provides drinking and irrigation water to approximately one sixth of the world's population (Barnett et al., 2005). The initial
layering of the snowpack is impacted by the deposition of falling snow, windblown snow redistribution, or a combination of
the two (Nienow and Campbell, 2011). Further densification can occur due to compaction and metamorphic mechanisms,
alongside meltwater, percolation, and refreeze events (Prowse and Owens, 1984; Tuttle and Jacobs, 2019; El Oufir et al., 2021;





Colliander et al., 2023). Given these factors, key elements of snow density are the age of the snowpack, snow depth, and water content. Fresh snow can have a snow density of 0.05 – 0.07 g/cm$^3$ while fresh damp snow can range from 0.10 – 0.20 g/cm$^3$ (Muskett, 2012). In contrast, the snow density of older dry snow is roughly 0.35 – 0.40 g/cm$^3$ and for older wet snow is up to 0.50 g/cm$^3$ (Seibert et al., 2015). Very wet snow and firn, which is snow that failed to melt in the previous summer and did not turn into ice, can contain a snow density ranging from 0.40 – 0.80 g/cm$^3$ (Muskett, 2012; Arenson et al., 2021). Within the northern hemisphere, there is an immense variation in average snow density which ranges from 0.05 – 0.59 g/cm$^3$ with an overall long-term average snow density of 0.25±0.07 g/cm$^3$ (Zhao et al., 2023).

Despite the attainability of snow density classification, there are significant complexities with generating the estimated snow density alongside the related snow depth and snow water equivalent (SWE) over large areas and in challenging environments such as thick forests and mountainous terrain. Snow depth is simply the total depth of snow on the ground while SWE can be defined as the resulting depth of water produced from the complete melt of a mass of snow (Henkel et al., 2018). The quantity of SWE is determined by the amount of snow accumulation alongside the amount of snow melt and sublimation (Xu et al., 2019). Field-based SWE datasets are both spatially and temporally scarce and can be expensive and labor intensive to acquire (Henkel et al., 2018; Fontrodona-Bach et al., 2023). In contrast, field-acquired snow depth measurements are more common, and are both easier and faster to obtain, though their spatial extent is also limited and can be challenging to obtain in difficult or remote areas (Collados-Lara et al., 2020; Tanniru and Ramsankaran, 2023). Automated stations can be utilized to collect snow measurements, which are rapidly becoming more commonplace, such as accounting for over 80% of the snow depth observing network north of 55° N in Canada (Brown et al., 2021). However, such stations may sometimes be primarily intended for non-climatic purposes such as for avalanche warnings and thus not be verified nor corrected for climatic trends (Salzmann et al., 2014).

Alternatives to field-based methods of snow observations are the use of airborne and spaceborne sensors to estimate snow properties which have achieved great success in recent decades (Nagler and Rott, 2000; Kelly et al., 2003; Marti et al., 2016; Cimoli et al., 2017; Tsai et al., 2019). Such sensors achieve large spatial coverage and the ability to clearly differentiate between snow and non-snow features (Nolin, 2010; Raghubanshi et al., 2023). However, many commonly used spaceborne sensors such as with the Landsat series, the Moderate Resolution Imaging Spectroradiometer (MODIS), the Advanced Very High Resolution Radiometer (AVHRR), and the Advanced Microwave Scanning Radiometer (AMSR-E/AMSR2) have limitations. These are either not capable of directly estimating snow depth or SWE, or, if able, have limited penetration or contain very coarse resolutions that make local scale estimation unattainable, in addition to potential cloud cover contamination (Rodell and Houser, 2004; Green et al., 2012; Lu et al., 2022; Stillinger et al., 2023). Repeat images captured via airborne Light Detection and Ranging (LiDAR) can serve to successfully estimate changes in snow depth (Deems et al., 2013; King et al., 2023); however the flights needed for these are costly, weather dependent, and require trained pilots and LiDAR specialists (Jacobs et al., 2021; Yu et al., 2022). While issues are present in relying solely on remote sensing for snow depth and SWE estimation, a blending of remote sensing imagery and field-based snow data can serve to significantly improve snow depth and SWE estimations (Kongoli et al., 2019; Pulliainen et al., 2020; Cammalleri et al., 2022; Venäläinen et al., 2023).



In addition to this, the inclusion of machine learning can expand the potential to estimate snow depth and SWE over
spatial and temporal scales. Machine learning techniques have been successfully applied to predict such features across Earth,
including high altitude and high latitude environments (Jonas et al., 2009; Zhang et al., 2021; Hu et al., 2023). Commonly
employed algorithms including Artificial Neural Network (ANN), K-Nearest Neighbor (KNN), Multiple Linear Regression
(MLR), Random Forest (RF), and Support Vector Machine (SVM) have achieved success in snow depth, SWE, and snow-
liquid ratio estimations (Broxton et al., 2019; Douglas and Zhang, 2021; Ntokas et al., 2021; Hoopes et al., 2023). Individually
many of these algorithms can produce positive results, though there may be a tendency for disagreement in model accuracy
and outcomes (Li et al., 2023). As an alternative, a weighted ensemble-based empirical model can be utilized to potentially
increase model accuracy, while also reducing estimation error (Douglas and Zhang, 2021; Brodylo et al., 2024). As each
algorithm is optimized differently to generate outputs, each containing their pros and cons, an ensemble approach can improve
feature estimation to ensure optimal results (Pes, 2020). A combination of such machine learning models, remote sensing
imagery, and field-based snow data can thus provide the necessary foundations to map snow features across the cryosphere,
which has been experiencing rising temperatures and increasing climatic uncertainty (Pan et al., 2017; Yang et al., 2020; Santi
et al., 2022).
One region where application of such a technique is worthwhile is in northern Europe, particularly in the Lapland
region located largely within the Arctic Circle. The area around Sodankylä, Finland is prone to long, cold winters with abundant
snowfall and both on-the-ground snow depth and SWE measurements are available for multiple months or more. Here, we
sought to utilize an object-based machine learning ensemble approach with a combination of time-series field and automated
snow data, alongside WorldView-2 (WV-2) imagery and LiDAR data to upscale snow depth, SWE, and snow density to a 10
$km^2$ local scale. This was implemented over six instances from December 2022 to April 2023, with snow estimates matched
to dominant vegetative communities. Field-based snow depth observations were upscaled first, before utilizing the estimated
snow depth to aid in upscaling more limited SWE field data to the local scale, with snow density then being mapped. Distinctive
machine learning algorithms were employed and compared to an ensemble-based technique for both snow depth and SWE
estimation.

## 2 Study area and data

### 2.1 Study area

The study area is found near the town of Sodankylä in the Sodankylä municipality of northern Finland, which is
roughly 125 km north of the Arctic Circle. The 10 $km^2$ site is located along the Kitinen River and hosts the Finnish
Meteorological Institute Arctic Space Centre (FMI-ARC) and the Sodankylä Geophysical Observatory (Bösinger, 2021)
between 67.356° N, 26.609° E, and 67.381° N, 26.693° E (Fig 1). It is largely flat, with elevations ranging between 170 and
190 m above sea level. Landcover consists primarily of coniferous and deciduous dominated forests and peat bogs, contains
organic and mineral soils, and portrays a standard flat northern boreal forest/taiga setting (Rautiainen et al., 2014). Field



analysis revealed a multitude of vegetative species at the study site. Dominant tree species are *Betula pubescens* (downy birch)
and *Pinus sylvestris* (Scots pine). Common shrub species include *Andromeda polifolia* (bog rosemary), *Empetrum nigrum*
(crowberry), *Rhododendron tomentosum* (Labrador tea), *Vaccinium cespitosum* (dwarf bilberry), *Vaccinium myrtillus*
(bilberry), *Vaccinium oxycoccus* (cranberry), and *Vaccinium vitis-idaea* (lingonberry). Graminoid species were comprised of
*Carex lasiocarpa* (woollyfruit sedge), *Danthonia decumbens* (heath grass), *Eriophorum vaginatum* (tussock cottongrass),
*Scheuchzeria palustris* (pod grass), and *Trichophorum cespitosum* (tufted bulrush). Forb species include *Comarum palustre*
(purple marshlock) and *Menyanthes trifoliata* (bog bean). Lichen and moss are also common.

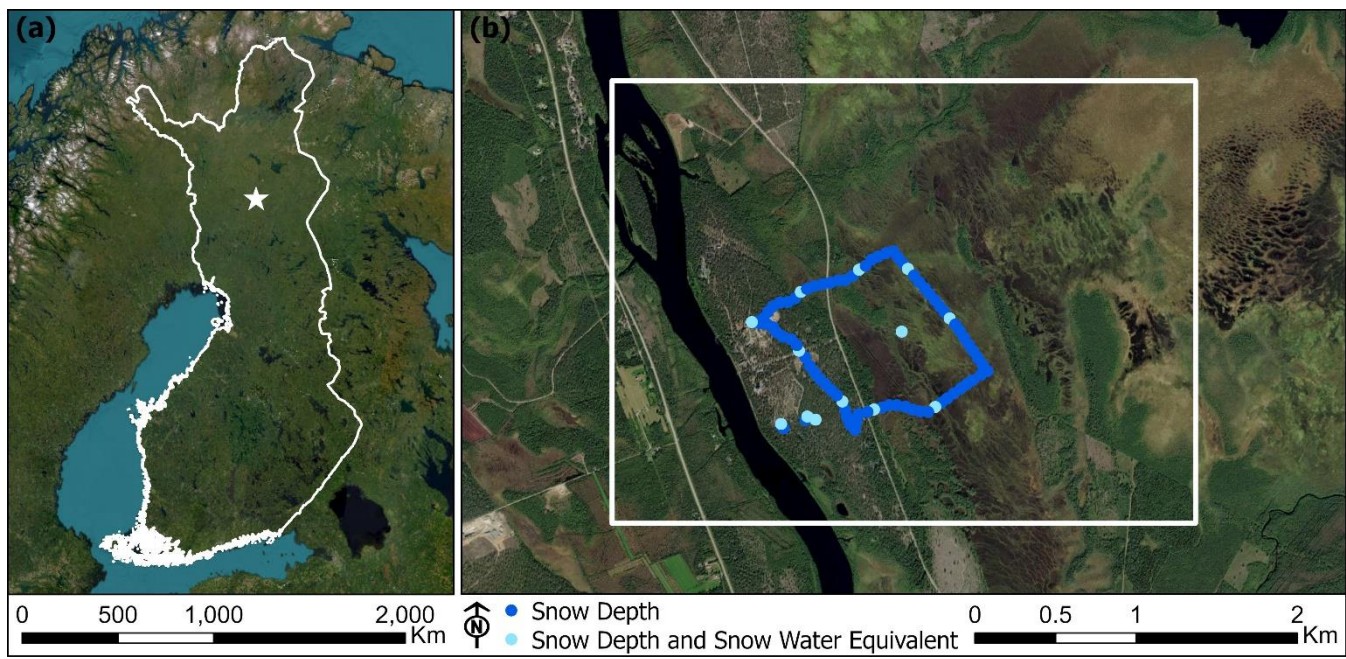

**Figure 1: Study area (a) in Sodankylä, Finland and (b) automated and manual snow depth and snow water equivalent measurements**
**within the 10 km² local scale study site. Image credits: © Esri, Earthstar Geographics, and Maxar.**

The climate in Sodankylä is defined by short but relatively warm summer season and a long and cold winter, with
snow present from October to May. Taiga snow is dominant, with thick layering of depth hoar at the base of the snowpack
(Anttila et al., 2014). Meaningful rain-on-snow events occur in November and early December (Bartsch et al., 2023). Between
1991 and 2020 at the FMI Sodankylä Tähtelä weather station, the average yearly precipitation was 543 mm with an average
yearly maximum snow depth of 91 cm that ranged from 65 – 127 cm. The average air temperature was 0.4 °C, the average
minimum was -4.2 °C, and the average maximum was 4.8 °C. The absolute minimum temperature was -49.5 °C while the
absolute maximum was 32.1 °C. The mean annual air temperature has increased by 0.07 °C from 2000 – 2018 (Bai et al.,
2021) and is expected to continue. Between the winters of 2007/08 to 2013/14 around FMI-ARC and the Sodankylä
Geophysical Observatory, the maximum SWE ranged approximately from 150 – 250 mm (Essery et al., 2016). For the winter



of 2022/23, a maximum snow depth of 99 cm was recorded at the Sodankylä Tähtelä weather station on 31 March 2023, with
rapid snow melt in April and early May (Fig 2). The average air temperature was generally near or below freezing in winter
and contained relatively low precipitation. The site generally contains low wind speeds that limit windblown snow
redistribution, with a monthly average of 2.5 – 2.9 m s$^{-1}$ above the forest canopy (Meinander et al., 2020).

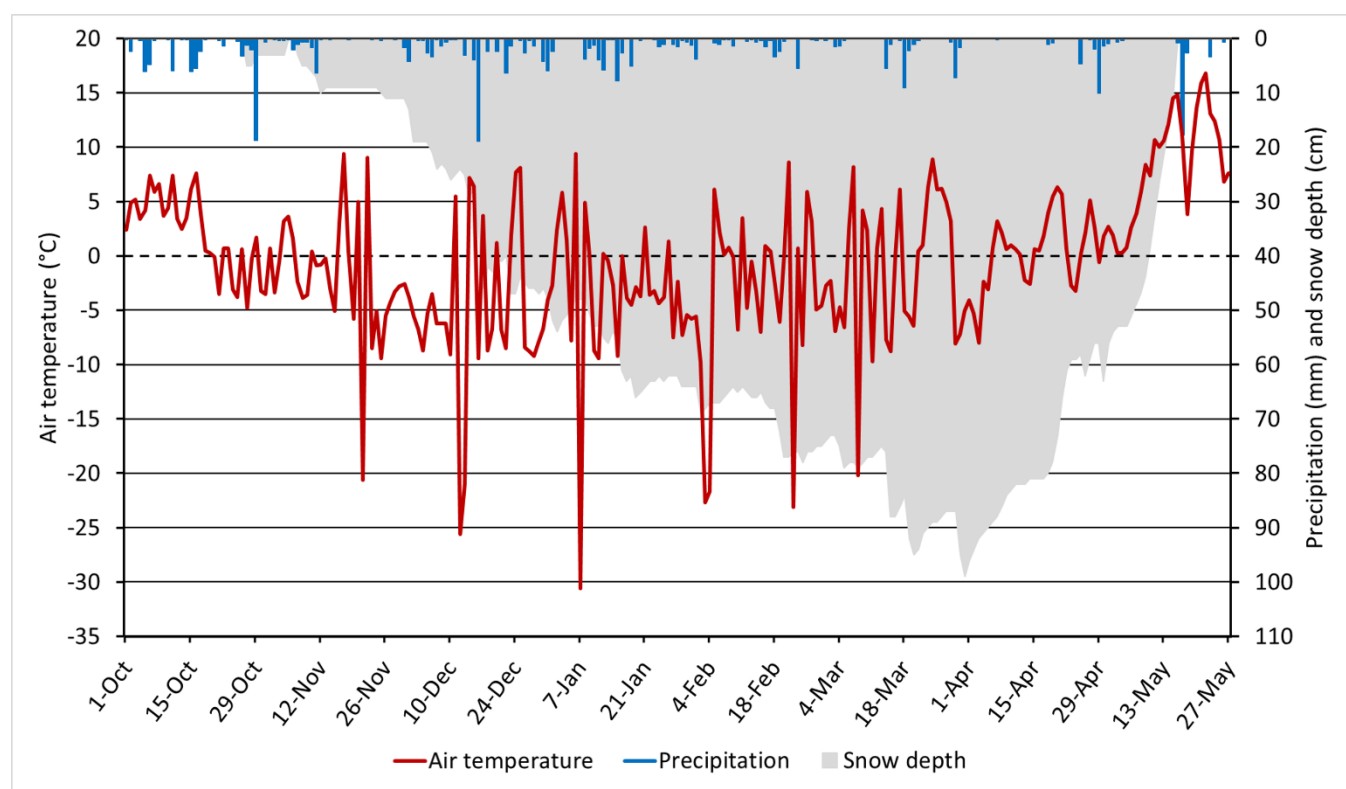


**Figure 2: Daily average air temperature (°C), precipitation (mm), and snow depth (cm) from the FMI Sodankylä Tähtelä weather**
**station from 01 October 2022 – 27 May 2023.**

## 2.2 Ground-based and remotely sensed measurements

Field-based snow data were acquired over distinct vegetative communities on 14 December 2022, 17 January 2023,
15 February 2023, 17 March 2023, 17 April 2023, and 28 April 2023. Manually obtained snow depth was measured with a
fixed stake or manual probe, while SWE was calculated with a scale that is paired to a snow tube that is 70 cm high and 10 cm
in diameter that includes a scale on the outside to measure snow depth (Leppänen et al., 2016). Automated observations were
performed for snow depth with the Campbell Scientific SR50 sonic distance instrument and for SWE with the Sommer
Messtechnik SSG 1000 snow scale instrument. A total of 88 repeat snowpack depth (cm) measurements were taken at the
same locations with 80 being manually recorded and 8 being acquired from automated stations (Fig 1(b)). Of these same 88
locations, a total of 13 repeat SWE (mm) measurements were recorded: 11 manually and 2 from automated stations. SWE



values were based on the total snowpack depth. An average daily value was recorded from the automated stations to match
with the field-based observations, with previously strong correlations found between the automated and manual measurements
for both snow depth and SWE with average correlation coefficients of 0.98 and 0.99, respectively (Leppänen et al., 2018).
Snow density (g/cm$^3$) was calculated from dividing SWE by snow depth at the same location.

On-the-ground vegetation data were acquired between 31 July and 4 August 2023. Plots were established randomly

along the snow depth measurement route to encompass major plant community types, primarily coniferous and hardwood
forests, and forested and herbaceous bogs. At each plot, a center point was established, flags were placed in each cardinal
direction to create a circular plot with a 7.3 m radius, and GPS coordinates of the center point and flags were recorded. In each
plot, all trees with diameter at breast height (DBH) greater than 10 cm were recorded by species and DBH. Five 0.5 m$^2$ quadrats
were randomly placed in each plot quadrant and aerial cover of the understory vegetation was estimated in 5% increments for
the following functional groups: moss, lichen, shrub, forb, and graminoid.

Cloud free and high spatial resolution (2 m) spaceborne WV-2 images were acquired on 02 August 2021 and 27 April

2023. The summer imagery contained spectral readings that matched with distinct vegetative communities, while the winter
imagery served to identify snow and non-snow features. Snow-free LiDAR data from 2020 was gathered from the National
Land Survey of Finland at a density of 5 pulses/m². Airborne LiDAR data were obtained on 27 April and 11 May 2023 by
NV5 Geospatial and contained full to partial snow cover. This was captured with a Leica City Mapper-2/Hypersion 2+ system
containing an average pulse density of $\geq$ 25 pulses/m$^2$, absolute vertical accuracy of $\leq$ 6 cm, relative vertical accuracy of $\leq$ 15
cm, and horizontal accuracy of $\leq$ 14 cm. The LiDAR data were further separated into a Digital Terrain Model (DTM), Digital
Surface Model (DSM), and Canopy Height Model (CHM). No major landcover changes impacted the study site during these
time periods that would have necessitated the need for repeat sets of imagery.

Land Use Land Cover (LULC) data were acquired from CORINE (Coordination of Information on the Environment)

Land Cover (CLC) at 20 m resolution from 2018. CLC is a LULC monitoring program that is coordinated by the European
Environment Agency (EEA) and is a current product of the Copernicus Land Monitoring Service (Aune-Lundberg and Strand,
2021). The LULC data was utilized to link vegetative communities to snow depth and SWE in the study area, while excluding
artificial features and water bodies. We downscaled the dataset to match the 2 m resolution WV-2 imagery and then updated
land cover boundaries where there were evident differences with the obtained summer imagery, thereby providing an updated,
higher-resolution LULC. In addition, a modified classification scheme was employed that sought to separate forest
communities by soil type and wetlands by moisture content. A RF-based classification scheme was employed that achieved
an Overall Accuracy (OA) of 91.7% and a Kappa value of 0.91, which indicated high LULC classification accuracy.





## 3 Methodology

### 3.1 Image segmentation

An Object-Based Image Analysis (OBIA) technique was utilized to make estimations of snow depth and SWE at the 10 km² local site scale In OBIA an image is separated into homogeneous groups of pixels known as image objects or segments, which are then utilized as the spatial unit for image assessment (Ye et al., 2018). This contrasts with more traditional pixel-based classification methods, in which image assessment is performed on a pixel-by-pixel basis. The OBIA approach was selected as it has been found to deliver enhanced accuracy and results over traditional pixel-based approaches, especially with high-resolution imagery (Sibaruddin et al., 2018; Shayeganpour et al., 2021; Ez-zahouani et al., 2023). Additionally, outputs generated from traditional pixel-based approaches can be susceptible to high local spatial heterogeneity between adjacent pixels, commonly known as the "salt-and-pepper" effect, which is not evident with OBIA (Wang et al., 2020).

Image segmentation was accomplished with the Segment Mean Shift tool in ArcGIS Pro software, a desktop GIS application. It contains a nonparametric iterative technique that utilizes kernel density estimation to generate image objects from a maximum of three image bands by grouping nearby pixels that contain similar spectral characteristics (Goldberg et al., 2021). The red, green, and near-infrared bands were utilized from the summer WV-2 imagery to carry out image segmentation. For parameters, the spectral detail was set to 19 (near maximum) while spatial detail was set to 1 (minimum) to improve segmentation as both heterogeneous and homogenous areas were present. A total of 37,917 unique image objects were created. Mean and standard deviation were calculated for each image object from the LiDAR and WV-2 datasets. Additional indices utilized included the Green Chlorophyll Index (GCI), Red-Edge Chlorophyll Index (RECI), Normalized Difference Vegetation Index (NDVI), Normalized Difference Water Index (NDWI), and Soil-Adjusted Vegetation Index (SAVI). Descriptions of these widely utilized indices, beyond the scope of this work, are available in Gaitán et al. (2013), Xue and Su. (2017), and Nadjla et al. (2022). The automated and field-based snow depth and SWE measurements were spatially joined to the generated image objects. In segments that contained two or more measurements, an average value was recorded.

### 3.2 Machine learning models

Commonly utilized and unique supervised regression-based machine learning models entailing of Random Forest (RF), Support Vector Machine (SVM), Artificial Neural Network (ANN), and Multiple Linear Regression (MLR) were chosen to estimate snow depth and SWE for the image objects. RF works by training a large collection of decision trees to generate an optimal output (Hwang et al., 2023). In contrast, SVM relies on an optimal hyperplane that minimizes error bounds (Pimentel et al., 2021). ANN is based upon the association of connected neurons like that of the human nervous system (Goel et al., 2023). MLR models the linear relationship between independent variables to a dependent variable (Kim et al., 2020). To aid in reducing potential modeling bias and overfitting, a $k$-fold cross-validation technique was employed. With this, matched data samples are randomly split into $k$ number of subsets, with $k-1$ being utilized to train models and the remainder





to test models (Abriha et al., 2023). Here, a *k*-fold of 10 was utilized whereby in each subset 90% of the data is assigned for
training and 10% is for testing, with output metrics determined from the average of all iterations.

**3.3 Object-based ensemble machine learning**

An object-based ensemble machine learning approach was applied from a combined weighted output of the RF, SVM,
ANN, and MLR models which is referred to here as Ensemble Analysis (EA). Given that these individual models compute
predictions differently and will have varying accuracies and errors, EA can result in a more robust model that considers more
accurate models while minimizing the influence of less accurate ones. This is relevant for repeat predictions over the same
study site as a model may perform well in one scenario while underperforming in another, such as with estimating snow depth
during a period of low or high snowfall. All four models were included to estimate snow depth, while SVM was dropped for
SWE estimation due to poor modeling results. The model weights for EA were determined by the correlation coefficient ($r$),
in which a model with a larger $r$ value would be given a higher weight, and the sum of weights equal to 1.0 (Zhang et al.,
2020). Combined model uncertainty for EA predictions was based on the standard deviation of model outputs and is referred
to as the standard deviation to ensemble prediction (STDE). Other statistical metrics included the Mean Absolute Error (MAE),
which is the absolute error between the observed and predicted values, and the Root Mean Square Error (RMSE), which is
more sensitive to outliers and is the square root of the mean squared error between observed and predicted values. Larger
differences between MAE and RMSE would serve to indicate a high variance of the individual errors from the test samples.
The $r$, MAE, and RMSE were calculated by:
$$r = \frac{\sum_{i=1}^{n}(p_i - \bar{p}_i)(o_i - \bar{o}_i)}{\sqrt{\sum_{i=1}^{n}(p_i - \bar{p}_i)^2}\sqrt{\sum_{i=1}^{n}(o_i - \bar{o}_i)^2}}, \tag{1}$$
$$MAE = \frac{\sum_{i=1}^{n}|p_i - o_i|}{n}, \tag{2}$$
$$RMSE = \sqrt{\frac{\sum_{i=1}^{n}(p_i - o_i)^2}{n}}, \tag{3}$$
where $n$ is the number of matched samples, $p_i$ is the model prediction, $o_i$ is the observed snow depth or SWE, $\bar{p}_i$ is the average
of the predicted values, and $\bar{o}_i$ is the average of the observed snow depth or SWE as adapted from Brodylo et al. (2024).
Local scale estimations were generated for snow depth via the ensemble-based approach, which were then utilized as
added inputs to aid in upscaling the more limited field acquired SWE data to the same local scale. Snow density was measured
by dividing the estimated SWE by the estimated snow depth in each respective instance. A summary of the methodology
framework can be found in Fig 3. Image objects were generated from multispectral imagery via image segmentation, with
averaged remote sensing and field snow depth values assigned to each unique image object (1). The spatially matched data
was then evaluated through the base machine learning models (RF, SVM, ANN, and MLR) to predict snow depth before being
ascertained with EA by combining model outputs with weighted averaging based on the $r$ value of each model (2). Model




metrics were obtained from each model alongside the mapped estimated local scale snow depth, with the estimated snow depth
from EA and field SWE values then being spatially joined to the previously matched input data (3). The updated spatially
matched data was analyzed by the base machine learning models (RF, ANN, and MLR) to predict SWE before being finalized
with EA (4). Model metrics were generated along with the mapped estimated local scale SWE in each instance (5).

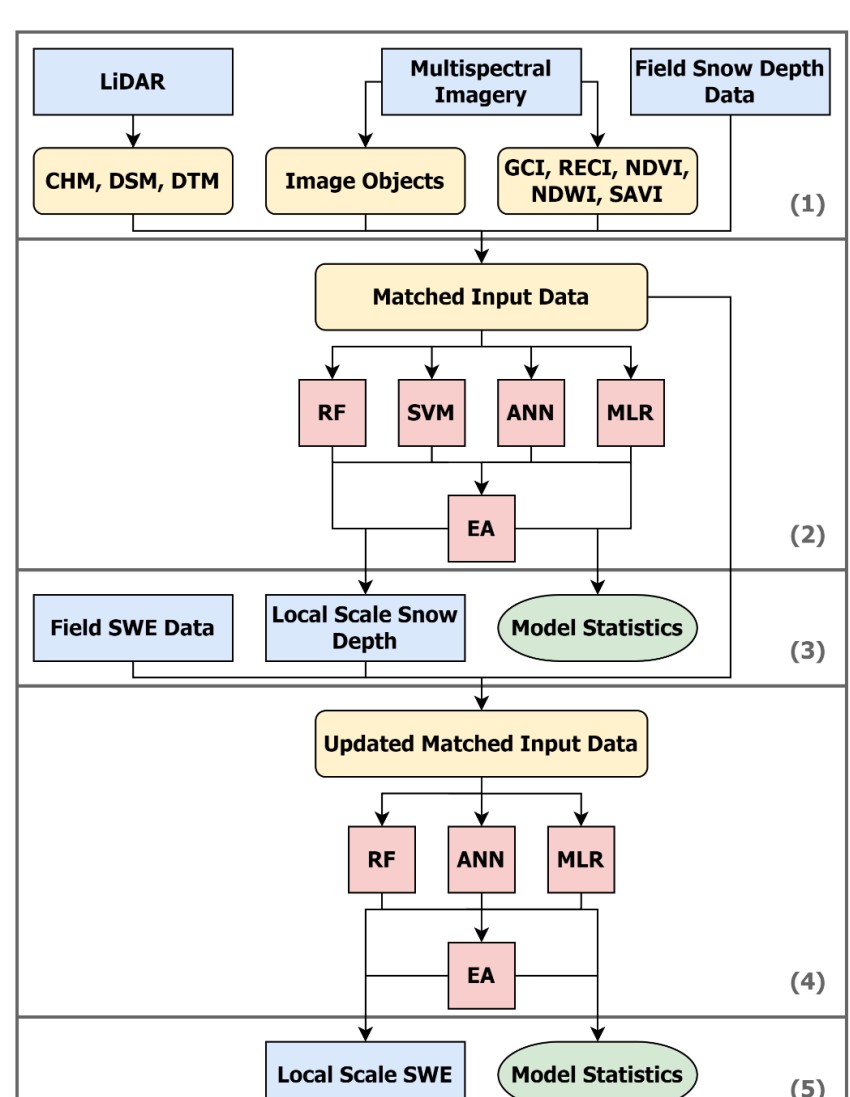

**Figure 3: Methodology framework to upscale field snow depth data to a local scale by using an object-based ensemble machine**
**learning approach, and then joining the produced snow depth outputs and matched input data with the field SWE data to generate**
**local scale SWE outputs. Blue indicates input/output data, yellow indicates processed data, red indicates machine learning, and**
**green indicates model metrics. RF is Random Forest, SVM is Support Vector Machine, ANN is Artificial Neural Network, MLR is**
**Multiple Linear Regression, and EA is Ensemble Analysis.**





While the methodology is similar to that found in Brodylo et al. (2024), that work was solely intent on upscaling 1
$m^2$ permafrost active layer thickness (ALT) field data to three 1 $km^2$ local scale sites in Alaska before then further upscaling
the ALT estimates to a 100 $km^2$ regional scale over multiple years. Here, we focused on first upscaling repeat field snow depth
measurements to a 10 $km^2$ local scale in Finland over multiple instances, and then combined the estimated snow depth data to
the original machine learning input data. The addition of snow depth as an input variable enabled a separate, enhanced estimate
of SWE at the same 10 $km^2$ local scale with more limited repeat field SWE measurements over the same multiple instances in
a single winter period. This then permitted snow density to be calculated at each moment in time from snow depth and SWE
estimations. The approach was applied to a shorter temporal analysis for snow depth, SWE, and snow density. It revealed how
each of these variables were interconnected during the initial, middle, and late winter, how machine learning models performed
over the course of the winter period, and how the studied variables related to landcover types over these different instances. In
addition, machine learning snow depth estimates were directly compared to independent LiDAR-based snow depth estimation.

## 4 Results

### 4.1 Snow depth

All tested models performed relatively well with the snow depth estimations. The best $r$, MAE, and RMSE values
were observed with EA in all instances (Table 1). March and April contained the highest $r$ values which were above or equal
to 0.67, and peaked with EA at 0.80 in March, 0.75 in early April, and 0.79 at the end of April. December, which had the
lowest snow depth, had the worst $r$ values with a minimum of 0.46 produced with ANN and a maximum of 0.63 produced
with EA. Owing to the lower snow depth, MAE and RMSE were the smallest out of all six instances at 2.8 cm and 3.6 cm for
EA, respectively. MAE and RMSE steadily increased for all models from roughly 2.8 – 3.3 cm and 3.6 – 4.4 cm in December
to 5.9 – 7.1 cm and 7.8 – 8.9 cm at the end of April. This was expected given increased snowfall and snow depth over time,
alongside minor periods of snowmelt throughout and accelerated snowmelt in April that would increase model uncertainty.
The base models of RF, SVM, ANN, and MLR generally contained similar values for each instance, with some variation. RF
and SVM performed well in all instances, though the former produced the worst $r$ (0.66) in January while the latter produced
consistently above average $r$, MAE, and RMSE values in all instances. ANN generated the poorest outcomes for $r$, MAE, and
RMSE (0.46, 3.3 cm, and 4.4 cm) in December, however outside of that it produced positive results, such as in January where
it produced the best for all three (0.69, 3.9 cm, and 4.8 cm). MLR performed on par with the other machine learning algorithms,
yet it arguably produced the poorest metrics, as it repeatedly delivered the highest MAE and RMSE in February (4.9 and 5.9
cm), March (4.8 and 6.0 cm), early April (6.4 and 8.4 cm), and the end of April (7.1 and 8.9 cm). More information about
outputs produced with EA for each instance can be seen in Fig 4, with each instance containing a 1:1 line, fitted linear
regression line, and scatterplot with STDE error bars in blue. With minor exceptions, there was largely an overall agreement
between the field and estimated snow depth values, and between the individual model outputs.





**Table 1: Machine learning model metrics for estimated snow depth with RF, SVM, ANN, MLR, and EA. MAE and RMSE are in**
**cm.**

| | 14-December-22 | | | | | | 17-January-23 | | | | |
|------|------|------|------|------|------|------|------|------|------|------|------|
| | RF | SVM | ANN | MLR | EA | | RF | SVM | ANN | MLR | EA |
| *r* | 0.60 | 0.61 | 0.46 | 0.56 | 0.63 | *r* | 0.66 | 0.69 | 0.69 | 0.68 | 0.73 |
| MAE | 2.9 | 2.8 | 3.3 | 3.2 | 2.8 | MAE | 4.1 | 4.0 | 3.9 | 4.0 | 3.7 |
| RMSE | 3.7 | 3.7 | 4.4 | 4.2 | 3.6 | RMSE | 4.9 | 4.9 | 4.8 | 5.1 | 4.5 |
| | 15-February-23 | | | | | | 17-March-23 | | | | |
| | RF | SVM | ANN | MLR | EA | | RF | SVM | ANN | MLR | EA |
| *r* | 0.73 | 0.73 | 0.67 | 0.62 | 0.73 | *r* | 0.78 | 0.78 | 0.70 | 0.72 | 0.80 |
| MAE | 3.9 | 3.7 | 4.1 | 4.9 | 3.6 | MAE | 4.4 | 4.0 | 4.6 | 4.8 | 3.9 |
| RMSE | 4.8 | 4.7 | 5.1 | 5.9 | 4.6 | RMSE | 5.4 | 5.1 | 5.7 | 6.0 | 4.9 |
| | 17-April-23 | | | | | | 28-April-23 | | | | |
| | RF | SVM | ANN | MLR | EA | | RF | SVM | ANN | MLR | EA |
| *r* | 0.67 | 0.68 | 0.68 | 0.67 | 0.75 | *r* | 0.76 | 0.74 | 0.74 | 0.73 | 0.79 |
| MAE | 5.7 | 5.7 | 5.8 | 6.4 | 5.1 | MAE | 6.1 | 6.4 | 6.6 | 7.1 | 5.9 |
| RMSE | 7.9 | 7.9 | 7.7 | 8.4 | 7.1 | RMSE | 8.2 | 8.5 | 8.5 | 8.9 | 7.8 |






**Figure 4: Scatterplot, 1:1 line (red line), and fitted regression line (black line) between the predicted snow depth from EA and the measured snow depth on each occasion from 14-December-2022 until 28-April-2023. STDE is in cyan.**

The snow depth average and standard deviation at each of the vegetative land cover types with the field data and local scale EA outputs are in Table 2. Mapped snow depth at the field scale and local scale estimates with EA for each instance from December 2022 – April 2023 can be seen in Fig 5. There was a general agreement and similar snow depth patterns in LULC's that contained both field and local scale data. The average snow depth was lowest for the field and local scale in December at 29 cm for both, while the highest readings were in March at 75 and 76 cm, with a rapid decline at the end of April at 36 and 37 cm. Standard deviation was lowest in December (±5 and ±3 cm) while highest at the end of April (±13 and ±8 cm) when there was increased snowmelt. At the field scale there was up to a 10 – 11 cm difference between coniferous forest (peat soil) and coniferous forest (mineral soil) from January to early April. The exception is at the end of April during the period of





snowmelt when field coniferous forest (mineral soil) contained higher snow depth at 43 cm than coniferous forest (peat soil)
at 40 cm. A similar pattern was evident with the field transitional woodland/shrub (peat soil) repeatedly containing higher
snow depths than transitional woodland/shrub (mineral soil) with a maximum difference of 10 cm in early April. However, at
the end of April both were equal at 36 cm of snow depth. Field-based peatbog (wet) and open area contained the lowest levels
of snow depth in all instances, ranging from 26 – 70 cm and 25 – 70 cm, respectively, with the latter experiencing elevated
standard deviation of ±20 and ±22 cm in the last two instances.



**Table 2: Mean and standard deviation (in parentheses) for snow depth (cm) estimates per LULC with field data and at the local scale with EA. Blank values indicate no field data.**

| | Snow depth field data | | | | | | Snow depth estimates with EA | | | | | |
|---|---|---|---|---|---|---|---|---|---|---|---|---|
| | 14-Dec-22 | 17-Jan-23 | 15-Feb-23 | 17-Mar-23 | 17-Apr-23 | 28-Apr-23 | 14-Dec-22 | 17-Jan-23 | 15-Feb-23 | 17-Mar-23 | 17-Apr-23 | 28-Apr-23 |
| Arable | | | | | | | 29 (3) | 60 (6) | 61 (6) | 77 (6) | 69 (5) | 37 (9) |
| Broad-leaved forest (mineral soil) | | | | | | | 31 (2) | 60 (3) | 60 (3) | 77 (4) | 71 (4) | 41 (6) |
| Broad-leaved forest (peat soil) | | | | | | | 32 (2) | 61 (3) | 61 (3) | 79 (3) | 70 (4) | 40 (6) |
| Coniferous forest (mineral soil) | 29 (5) | 55 (6) | 57 (4) | 74 (6) | 64 (7) | 43 (9) | 29 (2) | 57 (4) | 58 (3) | 75 (5) | 66 (4) | 40 (5) |
| Coniferous forest (peat soil) | 34 (2) | 66 (5) | 67 (5) | 84 (6) | 74 (7) | 40 (2) | 32 (1) | 63 (3) | 63 (2) | 81 (3) | 72 (3) | 42 (5) |
| Open area | 25 (4) | 54 (7) | 54 (6) | 70 (7) | 51 (20) | 33 (22) | 28 (3) | 60 (5) | 61 (4) | 77 (5) | 67 (6) | 39 (7) |
| Peatbog (dry) | | | | | | | 30 (2) | 59 (3) | 59 (3) | 73 (3) | 63 (5) | 36 (5) |
| Peatbog (wet) | 26 (4) | 52 (6) | 53 (8) | 70 (9) | 61 (10) | 27 (12) | 27 (2) | 56 (4) | 56 (4) | 73 (5) | 61 (6) | 30 (7) |
| Transitional woodland /shrub (mineral soil) | 29 (2) | 57 (6) | 57 (5) | 74 (7) | 60 (14) | 36 (14) | 30 (3) | 61 (4) | 61 (4) | 79 (5) | 70 (6) | 41 (7) |
| Transitional woodland /shrub (peat soil) | 31 (5) | 59 (6) | 61 (7) | 80 (6) | 70 (8) | 37 (9) | 30 (2) | 61 (4) | 61 (4) | 79 (5) | 69 (5) | 39 (7) |
| All LULC | 29 (5) | 56 (7) | 57 (7) | 75 (8) | 64 (11) | 36 (13) | 29 (3) | 58 (5) | 59 (4) | 76 (5) | 66 (7) | 37 (8) |




**Figure 5: Field and estimated snow depth (cm) in a) 14-December-22, b) 17-January-23, c) 15-February-23, d) 17-March-23, e) 17-April-23, and f) 28-April-23 alongside g) a LULC map based on data from CLC and h) 28-April-23 snow depth difference from 27-April-23 collected LiDAR.**

At peak snow depth at the local scale in March, both dry and wet peatbogs contained the lowest average snow depth at 73 cm. Dry, unsaturated peatbog was found to have snow depths equal to or greater than wet, saturated peatbog, with a





difference of 3 cm in the first three months, equal in March and early April, and then jumping to 6 cm at the end of April during more intense snow melt. Arable and open area contained similar estimated snow depth values in all instances and were higher than dry and wet peatbogs from January to the end of April. Forests and transitional woodlands largely contained the highest average values in March with broad-leaved forest recording 77 cm (mineral soil) and 79 cm (peat soil), coniferous forest (peat soil) with 81 cm, and transitional woodland/shrub containing 79 cm in both mineral and peat soil. There was also a consistent 1-2 cm snow depth difference between the local scale broad-leaved forest peat soils and mineral soils, with the former having higher snow depth leading up to peak snow depth in March, while the inverse was evident post peak snow depth. Transitional woodland/shrub mimicked this during post peak snow depth with a 1-2 cm snow depth difference between mineral and peat soil. Local scale coniferous forest (peat soil) consistently contained snow depth values greater than coniferous forest (mineral soil), with up to a 5 – 6 cm difference from January to early April. In addition, field and local scale snow depth estimates from 28 April were compared to the difference between snow covered DTM from the prior day and snow-free DTM from 2020. Results indicate field snow depth measurements generally exceeded the estimated LiDAR-based snow depth estimations by an average of 9.6 cm, while for the local scale with EA it was lower at 5.4 cm.

**4.2 Snow water equivalent**

Machine learning model performance for SWE estimation between RF, ANN, MLR, and EA can be seen in Table 3. Given more limited field-based SWE measurements with 13 samples, the models encountered more pronounced challenges matching estimations to real-world data yet were generally able to produce acceptable results. SVM was dropped due to poor performance in all instances. ANN, MLR, and EA contained relatively stable and positive metrics for $r$, MAE, and RMSE in all instances. EA generally produced the best metrics, although MLR performed best in some instances. Metrics from RF varied considerably, being on-par with the other models in December, February, and late April while poor in January, March, and early April. Despite this, RF was included in the weighted ensemble procedure given that in some instances it produced acceptable outcomes, while in others the low $r$ value would greatly minimize its weight. A scatterplot, 1:1 line, and fitted linear regression line for each instance of SWE predictions produced by EA alongside STDE can be seen in Fig 6. December contained the poorest metrics, with a maximum $r$ of 0.37 with MLR and EA, which may be connected to the poorer snow depth metrics in that same month, while March contained the highest $r$ of 0.87 with MLR and 0.79 with EA. Similarly with the snow depth metrics over the same period, MAE and RMSE were lowest in December from roughly 5.0 – 6.6 mm and 5.9 – 7.9 mm before rising to become the highest at the end of April at 24.1 – 33.4 mm and 33.4 – 41.5 mm.





**Table 3: Machine learning model metrics for estimated snow water equivalent with RF, ANN, MLR, and EA. MAE and RMSE are in mm.**

| | 14-December-22 | | | | | 17-January-23 | | | |
|---|---|---|---|---|---|---|---|---|---|
| | RF | ANN | MLR | EA | | RF | ANN | MLR | EA |
| $r$ | 0.30 | 0.36 | 0.37 | 0.37 | $r$ | 0.05 | 0.69 | 0.63 | 0.71 |
| MAE | 5.0 | 6.2 | 6.6 | 5.8 | MAE | 9.0 | 8.4 | 6.9 | 6.7 |
| RMSE | 5.9 | 7.5 | 7.9 | 6.6 | RMSE | 11.0 | 10.5 | 9.6 | 8.4 |
| | 15-February-23 | | | | | 17-March-23 | | | |
| | RF | ANN | MLR | EA | | RF | ANN | MLR | EA |
| $r$ | 0.71 | 0.67 | 0.70 | 0.72 | $r$ | 0.16 | 0.64 | 0.87 | 0.79 |
| MAE | 11.0 | 8.9 | 8.3 | 6.7 | MAE | 15.2 | 12.0 | 6.3 | 8.2 |
| RMSE | 12.3 | 11.6 | 10.9 | 9.8 | RMSE | 17.4 | 16.0 | 8.7 | 10.9 |
| | 17-April-23 | | | | | 28-April-23 | | | |
| | RF | ANN | MLR | EA | | RF | ANN | MLR | EA |
| $r$ | 0.09 | 0.70 | 0.72 | 0.73 | $r$ | 0.55 | 0.56 | 0.71 | 0.67 |
| MAE | 19.4 | 14.9 | 12.9 | 13.5 | MAE | 33.4 | 30.5 | 31.2 | 24.1 |
| RMSE | 22.5 | 18.0 | 16.0 | 15.8 | RMSE | 39.2 | 41.5 | 36.9 | 33.4 |







**Figure 6: Scatterplot, 1:1 line (red line), and fitted regression line (black line) between the predicted SWE from EA and the measured SWE on each occasion from 14-December-2022 until 28-April-2023. STDE is in cyan.**

The *r* values produced by RF ranged from a very poor correlation of 0.05 in January to a high correlation of 0.71 in February. Despite this, RF easily contained the best MAE and RMSE from all models in December (5.0 and 5.9 mm) alongside the best *r* in February from all base models at 0.71, with EA at 0.72. ANN primarily contained metrics that were intermediate, yet from the base models it was able to produce in January the best *r* at 0.69 and in late April the best MAE at 30.5. Out of the base models, MLR generally achieved the best *r*, MAE, and RMSE in most instances, with it being especially dominant in March (0.87, 6.3 mm, and 8.7 mm) and early April (0.72, 12.9 mm, and 16.0 mm). However, it also produced the worst MAE and RMSE in December (6.2 and 7.9 mm). In contrast, EA continually generated the best or second-best metrics in all instances for *r*, MAE, and RMSE. EA was particularly dominant with the best *r*, MAE, and RMSE in January (0.71, 6.7 mm, 8.4 mm) and February (0.72, 6.7 mm, 9.8 mm). This was despite the large variation in *r* for RF in both months (0.05 and 0.71). In





December it matched MLR for the highest $r$ (0.37), while in early April it contained slightly better $r$ (0.73) and RMSE (15.8).
At the end of April, EA generated the lowest MAE (24.1 mm) and RMSE (33.4 mm), which were notably better than the best
base model MAE from ANN (30.5 mm) and RMSE from MLR (36.9 mm).

The average and standard deviation of SWE field data and local scale EA outputs at the vegetative land cover types

for all instances can be seen in Table 4. There were particularly notable SWE disparities at the end of April between peatbog
and forest communities. As with snow depth, the average field and local scale SWE were lowest in December (34 and 35 mm),
while they were highest in early April (177 and 187 mm), post-peak snow depth. Standard deviation increased over the period
at both the field and local scale from ±6 and ±5 mm in December to ±46 and ±33 mm at the end of April. Across all instances
the field SWE for coniferous forest (mineral soil), open area, and peatbog (wet) ranged from 30 to 173 mm, 34 to 176 mm,
and 37 to 180 mm, respectively. Field SWE was repeatedly higher in transitional woodland/shrub (peat soil) ranging from 38
to 192 mm. The exception was at the end of April when the inverse occurred, and it recorded the lowest SWE (119 mm)
alongside the highest standard deviation of ±80 mm that was influenced by extreme SWE field data variation. At the local
scale, SWE continually ranged higher at coniferous forest (peat soil) from 36 to 191 mm than at coniferous forest (mineral
soil) from 29 to 185 mm, despite the lack of on-the-ground data for the former. At the end of April, the difference increased
to 19 mm. For both broad-leaved forest and transitional woodland/shrub, the opposite was found with SWE values tending to
be higher in mineral soil than in peat soil from January to the end of April. For broad-leaved forest the difference peaked at 6
mm at the end of April, while for transitional woodland/shrub it was up to 18 mm in both March and the end of April. Local
scale peatbog (dry) had higher or equal to average SWE than peatbog (wet) from January to end of April, ranging from 111 –
181 mm, compared to 108 – 178 mm. In late April the difference widened to 20 mm. Local scale arable and open area contained
continually higher SWE values than dry and wet peatbogs between January and late April. A distribution of SWE over the 10
$km^2$ site for each instance from December 2022 – April 2023 can be seen in Fig 7, which illustrates where and how much SWE
varied over time for the field data and EA-based local scale outputs.



**Table 4: Mean and standard deviation (in parentheses) for SWE (mm) estimates per LULC with field data and at the local scale with EA. Blank values indicate no field data.**

| | SWE field data | | | | | | SWE estimates with EA | | | | | |
|---|---|---|---|---|---|---|---|---|---|---|---|---|
| | 14-Dec-22 | 17-Jan-23 | 15-Feb-23 | 17-Mar-23 | 17-Apr-23 | 28-Apr-23 | 14-Dec-22 | 17-Jan-23 | 15-Feb-23 | 17-Mar-23 | 17-Apr-23 | 28-Apr-23 |
| Arable | | | | | | | 35 (7) | 123 (15) | 142 (8) | 183 (14) | 194 (11) | 149 (35) |
| Broad-leaved forest (mineral soil) | | | | | | | 35 (5) | 123 (7) | 139 (6) | 194 (11) | 193 (11) | 167 (22) |
| Broad-leaved forest (peat soil) | | | | | | | 36 (4) | 120 (8) | 140 (7) | 190 (11) | 191 (12) | 161 (24) |
| Coniferous forest (mineral soil) | 30 (7) | 105 (9) | 132 (17) | 159 (11) | 173 (19) | 140 (34) | 29 (6) | 120 (10) | 134 (7) | 180 (14) | 185 (14) | 150 (24) |
| Coniferous forest (peat soil) | | | | | | | 37 (3) | 123 (5) | 143 (4) | 192 (8) | 198 (8) | 169 (17) |
| Open area | 34 (9) | 115 (12) | 136 (22) | 164 (26) | 176 (44) | 142 (55) | 37 (5) | 128 (11) | 140 (6) | 193 (19) | 194 (14) | 164 (24) |
| Peatbog (dry) | | | | | | | 36 (2) | 111 (8) | 133 (6) | 169 (12) | 181 (12) | 133 (25) |
| Peatbog (wet) | 37 (3) | 107 (13) | 131 (14) | 162 (21) | 180 (17) | 121 (45) | 37 (2) | 108 (10) | 133 (8) | 165 (17) | 178 (15) | 113 (29) |
| Transitional woodland /shrub (mineral soil) | | | | | | | 36 (6) | 130 (9) | 142 (6) | 201 (12) | 198 (12) | 172 (22) |
| Transitional woodland /shrub (peat soil) | 38 (5) | 120 (11) | 137 (14) | 183 (21) | 182 (35) | 119 (80) | 38 (2) | 117 (9) | 141 (7) | 183 (14) | 194 (11) | 154 (27) |
| All LULC | 34 (6) | 110 (11) | 133 (15) | 166 (18) | 177 (23) | 131 (46) | 35 (5) | 117 (11) | 137 (8) | 180 (18) | 187 (15) | 144 (33) |





**Figure 7: Field and estimated SWE (mm) in a) 14-December-22, b) 17-January-23, c) 15-February-23, d) 17-March-23, e) 17-April-23, and f) 28-April-23 alongside g) a LULC map based on data from CLC.**

## 4.3 Snow density

Snow density is the ratio between the volume of water produced by melting a given volume of snow and the original volume of snow itself. This percentage refers to the water content within a given volume of snow. In general, fresh snowfall has low density while older, compacted, or wind-effected snow will have a higher density. Table 5 contains the mean and standard deviation of the snow density percentage for each vegetative landcover type from December to the end of April. The average snow density percentage for field and local scale data was lowest in December with 12% for both, while the highest was at the end of April at 36% and 39%, respectively. Standard deviation for the combined averages were generally low, with



a maximum of ±4% and ±5% in late April for field and local scale EA estimates. For the first five instances, snow density
percentages were slightly higher with the canopy-free open area and peatbog (wet), which ranged from 14 – 31% and 13 –
29%. In contrast, the more tree-covered coniferous forest (mineral soil) and transitional woodland/shrub (peat soil) routinely
experienced lower percentages ranging from 11 – 27% and 13 – 27%. In the final instance, field transitional woodland/shrub
(peat soil) and peatbog (wet) had the highest snow density percentages at 42% and 39%, while open area and coniferous forest
(mineral soil) were markedly lower at 33% and 32%.

**Table 5**: **Mean and standard deviation (in parentheses) for snow-to-water-percentage estimates per LULC with field data and EA.**
**Blank values indicate no field data.**

| | Snow-water-percentage field data | | | | | | Snow-water-percentage estimates with EA | | | | | |
|---|---|---|---|---|---|---|---|---|---|---|---|---|
| | 14-Dec-22 | 17-Jan-23 | 15-Feb-23 | 17-Mar-23 | 17-Apr-23 | 28-Apr-23 | 14-Dec-22 | 17-Jan-23 | 15-Feb-23 | 17-Mar-23 | 17-Apr-23 | 28-Apr-23 |
| Arable | | | | | | | 12 (2) | 20 (2) | 23 (2) | 24 (2) | 29 (2) | 42 (8) |
| Broad-leaved forest (mineral soil) | | | | | | | 11 (1) | 21 (2) | 23 (1) | 25 (2) | 27 (2) | 41 (7) |
| Broad-leaved forest (peat soil) | | | | | | | 11 (1) | 20 (2) | 23 (1) | 24 (2) | 27 (1) | 41 (5) |
| Coniferous forest (mineral soil) | 11 (3) | 20 (1) | 23 (3) | 22 (1) | 27 (1) | 32 (1) | 10 (2) | 21 (2) | 23 (1) | 24 (3) | 28 (2) | 38 (5) |
| Coniferous forest (peat soil) | | | | | | | 12 (1) | 20 (1) | 23 (1) | 24 (1) | 28 (1) | 41 (3) |
| Open area | 14 (1) | 21 (1) | 24 (1) | 22 (1) | 31 (5) | 33 (1) | 13 (2) | 21 (2) | 23 (2) | 25 (2) | 29 (2) | 42.5 (5) |
| Peatbog (dry) | | | | | | | 12 (1) | 19 (1) | 23 (1) | 23 (1) | 29 (1) | 37 (4) |
| Peatbog (wet) | 13 (1) | 21 (2) | 23 (1) | 22 (1) | 29 (2) | 39 (1) | 14 (1) | 19 (1) | 24 (1) | 23 (1) | 30 (1) | 38 (5) |
| Transitional woodland /shrub (mineral soil) | | | | | | | 12 (2) | 21 (2) | 23 (1) | 26 (2) | 28 (2) | 43 (6) |
| Transitional woodland /shrub (peat soil) | 13 (2) | 20 (1) | 23 (1) | 23 (1) | 27 (1) | 42 (1) | 13 (1) | 19 (1) | 23 (1) | 23 (1) | 28 (1) | 40 (4) |





| | 12 | 20 | 23 | 22 | 28 | 36 | 12 | 20 | 23 | 24 | 28 | 39 |
|---|---|---|---|---|---|---|---|---|---|---|---|---|
| All LULC | (2) | (1) | (2) | (1) | (3) | (4) | (2) | (2) | (1) | (2) | (2) | (5) |

As with the field averages, for the local scale averages from December to early April there were generally minimal differences in snow density between different land cover types while experiencing greater fluctuations at the end of April with a maximum difference of 6%. Local scale arable and open area contained the same averages in three instances with open area also having a 1% higher increase in snow density in three instances. Peatbog (wet) contained percentages equal or up to 2% higher than peatbog (dry) in all instances. Average snow density percentage on transitional woodland/shrub (mineral soil) was equal to or up to 3% higher than for peat soil from January to the end of April, with broad-leaved forest (mineral soil) being equal to the broad-leaved forest (peat soil) in four instances and up to 1% higher in the remaining two. For coniferous forest it was relatively stable between the mineral and peat soils until the end of April when the average was 38% for mineral soil and 41% for peat soil. At the end of April for the local scale, the lowest snow density averages were recorded for dry and wet peatbogs at 37% and 38%, alongside 38% for coniferous forest (mineral soil). In contrast, the highest average snow density percentages at the local scale were in transitional woodland/shrub (mineral soil) at 43%, along with both arable and open area at 42%. A spatial view of the gradual increase in the snow density percentage across the six instances with the rapid rise at the end of April can be seen in Fig 8.





**Figure 8: Field and estimated snow density percentage in a) 14-December-22, b) 17-January-23, c) 15-February-23, d) 17-March-23, e) 17-April-23, and f) 28-April-23 alongside g) a LULC map based on data from CLC.**

## 5 Discussion

With snow depth estimation, all models performed well, with EA generating the best statistics. As is common for the study region the snow depth was lowest in December and highest in March before daily temperatures began exceeding 0 °C in April. There were consistent differences in snow depth between different vegetative communities. This was most apparent with higher snow depth being associated with broad-leaved forests, transitional woodland/shrubs, and particularly with coniferous forest (peat soil). Shallower snow depth was recorded at arable, coniferous forest (mineral soil), open areas, and both dry and wet peatbogs. With peatbogs, wet peat conducts heat better than dry peat resulting in heat flowing more effortlessly in wet peat layers in winter (Kujala et al., 2008), which may result in increased snowmelt and compaction.



Furthermore, mineral soil is more thermally conductive than peat soil (Atchley et al., 2016), which may promote snowmelt
and compaction in similar vegetation communities containing mineral soil compared with peat soil where snowmelt and
compaction would be reduced. Forests with drier mineral soils were generally more shielded from saturated soil found in
peatbogs, while forests with peat soil were oftentimes adjacent to peatbogs. As the water table in many parts was at or near the
surface, adjacent soils would contain greater soil saturation while the shielded mineral soils would in theory be more
unsaturated. A notable exception is for approximately half of the broad-leaved forest (mineral soil) that is along the Kitinen
River, which may have especially influenced snow depth, SWE, and snow density readings for that LULC. Given that saturated
soil needs greater energy to heat than does unsaturated soil (Howe and Smith, 2021), saturated soil would require greater
energy to warm in the spring and remain warmer in the winter than the unsaturated soil, which would have a resulting impact
on snow cover. Post winter soil thaw varied with five FMI Campbell Scientific 109-L soil temperature sensors in the study
area at 5 and 10 cm below the surface. For two sensors found in coniferous forest and one in an open area with mineral soil,
the soil fully thawed out between 10 – 25 April, while for the two sensors in the peatland, the soil thawed out from 11 – 13
May, which would have aided in accelerating overlaying snow cover melt for the former. It should be noted the impact that
direct solar radiation may have on the energy balance of the snowpack and melt processes, along with wind impacted (open
areas) versus wind protected (forest) vegetative communities. Lastly, snow interception and sublimation are major factors in
forest communities, especially with conifers, which can lead to a notable diversity of snow accumulation on the forest floor
(Helbig, 2020).
For the SWE estimations, model results were more mixed, but nonetheless promising. ANN, MLR, and EA were all
able to produce encouraging metrics, while there was elevated variation with RF. EA consistently produced the best or second-
best metrics, and generally produced the best metrics. MLR also performed well despite being the simplest form of machine
learning in this study. In comparison to the snow depth there was a much smaller sample size which led to greater model
uncertainty and disagreement. A greater number of SWE field samples would have provided enhanced findings; however,
these field measurements can be time-consuming and expensive to collect across a large geographic region, with SWE
measurements taking approximately 20 times as long to complete compared to snow depth measurements (Sturm et al., 2010).
Nonetheless SWE was found to be lowest in December and highest in early April, which was post-peak snow depth. With the
field data, it was found that SWE was higher in transitional woodland/shrub (peat soil) than with coniferous forest (mineral
soil), which may be attributed to potentially more saturated peat soil allowing for greater water retention within the snow
cover, while the unsaturated mineral soil drained slightly more liquid from the overlaying snow cover. Mineral soils across the
study site are sand-rich and would be dry most of the time at the surface and likely never reach saturation, with any melted
snow being drained in these soils. The one exception was with the end of April when there was a notable reversal, which may
have been due to increased snow interception, snowmelt, sublimation, and windblown snow from branches in some vegetation
types. A similar trend was observed at the local scale. Local scale coniferous forest (peat soil) continually contained higher
average SWE than coniferous forest (mineral soil) which may be the result of the unsaturated mineral soil absorbing water
from the overlaying snow while the saturated peat soil slowed the draining of water through the snowpack and into the soils.





Wet and dry peatbogs largely contained the lowest SWE measurements. These low open areas likely experienced enhanced
wind activity that blew snow laterally away while also leading to greater sublimation. This would have led to greater snow
particle cohesion and denser wind slab layer formation at the surface of the snowpack due to sintering after snow was mobilized
in the wind (Mott et al., 2018).

Lastly, snow density was lowest in December and increased until the end of April when it was highest, which was

during a period of rapid snowmelt. This was to be expected given that the beginning and middle winter typically contain larger
quantities of fresh snowfall, while by the end of winter the snowpack would have compacted over time and become denser as
the snowpack reaches an equilibrium temperature state of 0 °C (e.g., isothermal). As the snowpack develops, a larger snow
grain size (depth hoar) results in a lower density in shallow snowpack. However, as the snowpack becomes isothermal, the
depth hoar layer will metamorphose and become denser, especially near the ground (Gu et al., 2019). With the field data, a
higher snow density percentage was observed at the end of April in peatbog (wet) and transitional woodland/shrub (peat soil)
which contrasted with coniferous forest (mineral soil) and open area and may be attributed to soil saturation for those specific
locations. At the end of April for the local scale the highest snow density percentages were found in vegetative communities
that were more impacted by wind such as arable, open area, and transitional woodland/shrub (mineral soil) by a slight amount.
In contrast, coniferous forest (mineral soil) along with wet and dry peatbogs contained the lowest percentages with landcover
containing peat soil experiencing higher snow density percentages than with landcover containing mineral soil. Local scale
wet peatbog was found to generally contain slightly higher amounts than dry peatbog. This may be attributed to dry peatbog
being on average ~2.2 m higher in elevation than wet peatbog in our study area, which may have contributed to the movement
of water over time to wet peatbogs at incrementally lower elevations.

Solar radiation increased throughout the timeframe and was not uniform over the study area, such as with thick forests

sometimes obscuring adjacent canopy-free areas from solar radiation. As this would have impacted real-world snow estimates,
we incorporated end of winter WV-2 imagery in the framework as it was able to aid in capturing such irregularities. A limited
quantity and spatial extent of field measurements restricted further associations with vegetative communities, especially for
SWE and, in turn, snow density. Had additional measurements been taken at communities missing field data, there would be
a more comprehensive understanding of snow-landcover relationships. Additional datasets would have likely improved the
model statistics and estimation of all three studied features. Soil moisture and air/subsurface temperature data were accessible
in the study area yet were excluded, despite their strong association with snow depth and SWE (Contosta et al., 2016). This
was due to a limited number of these measurements that corresponded to the six instances, with some containing gaps or
missing data which would hinder spatial mapping and association with landcover types. Furthermore, very few of these
measurements were located on or adjacent to the field snow depth and SWE measurements, which severely limited a proper
linkage between the field data with soil moisture and temperature. Additional remote-sensing based data could have been
utilized as an add-on to assist in mapping soil moisture and temperature for the study, alongside improving estimations for
snow depth and SWE. However, due to the vegetative heterogeneity at the 10 km$^2$ site and clustering of the field data, medium
and low-resolution imagery would have provided questionable benefit. High-resolution hyperspectral imagery and Synthetic



Aperture Radar (SAR) are particularly relevant, given the additional available spectral bands of the former and the proven
application with snow depth and SWE detection in the latter (Patil et al., 2020), and would have likely benefited the findings.

## 6 Conclusions

We employed an object-based machine learning ensemble approach with time-series field snow depth and SWE data
in northern Finland to first estimate snow depth at a local scale, before incorporating the snow depth outputs to estimate SWE
at the same local scale alongside generating snow density estimations from six instances between December 2022 and April
2023. Snow depth peaked in March, SWE peaked shortly after in early April, and snow density peaked with the final available
data at the end of April. Multiple machine learning models, particularly with the ensemble approach, were shown to positively
estimate key snowpack attributes over the period at the study site in Sodankylä. We established that there are direct spatial and
temporal connections between three commonly studied snowpack elements with vegetation and soil types, with more research
recommended to further characterize these associations. Although there is promise with intricate machine learning techniques,
this study also highlights opportunities to assess where less complex methods may be employed for computational efficiency,
especially when scaling up. While performed over a small portion of northern Finland, when matched with other field-based
snowpack and remote sensing data across the region it would be possible to further upscale the studied snow-based estimates
over a wider, regional-scale over various periods in time. This would also need to account for differing types of snowpack,
terrain, and vegetative communities found throughout the pan-Arctic domain. As average temperatures around the Arctic are
projected to increase with fewer days below freezing, more uncertain climactic conditions and precipitation events would
affect the quantity, rate, and timing of snowfall, snow-on/snow-off, and snowmelt runoff in the region. Given that waterbodies
such as lakes, ponds, and rivers in Finland and other high latitude areas are fed by the annual snowmelt, any changes to this
natural process would meaningfully alter the hydrological makeup. The machine-learning based methodology applied in this
effort can serve to benefit future snow-related analyses in high latitude regions, alongside other areas on Earth that regularly
experience seasonal snow.
*Data availability.* Field snow depth and snow water equivalent data is maintained by the Finnish Meteorological Institute and
is available at https://litdb.fmi.fi/index.php. Additional study data is available upon reasonable request.
*Author contributions.* DB, LVB, EJD, and TAD designed and initiated the study. RRB classified vegetation. EJD obtained
LiDAR data. JL obtained field snow observations. DB, LVB, and EJD developed the methodology. DB wrote the initial draft
and figures. All authors contributed to manuscript development and review.
*Competing interests.* The authors declare that they have no conflict of interest.
*Acknowledgements.* Staff at the Finnish Meteorological Institute are acknowledged for providing field measurements.



*Financial support.* This research was funded by the US Department of Defense - PE 0602144A Program Increase 'Defense Resiliency Platform Against Extreme Cold Weather'.

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
