# Peer review of "Object-based ensemble estimation of snow depth and snow water 2 equivalent over multiple months in Sodankylä, Finland"

_EGUsphere, 2024_

## Author Comment (AC1)

Dear Reviewer #1,

We highly appreciate you taking your time to review our manuscript and provide valuable feedback to aid in addressing weak points and areas of concern, while also seeking to strengthen the novelty of the presented work. In the following, we address each of the points raised. Black text indicates the reviewer's comments. Blue text indicates our response and changes to the manuscript.

**General Comments:**

Brodylo et al.'s manuscript is well-written, structured clearly, and supported by strong graphical presentation, providing a straightforward exploration into snow depth and snow water equivalent (SWE) estimation using an ensemble machine learning approach. The integration of LiDAR, remote sensing imagery, and in-situ observations is logical and aligns well with the type of studies frequently published in this journal. However, I have several significant concerns regarding the novelty of the approach, methodological clarity, and the limited sample size—particularly for SWE estimation—that need to be thoroughly addressed before the paper can be considered for publication. I have outlined these major concerns, along with specific suggestions for improvement, in detail below.

**Major Comments:**

**1.** Currently, the paper's primary novel contributions are unclear to me. While the presented approach effectively integrates established practices (ensemble machine learning methods, LiDAR-based snow depth estimation), the methodological novelty seems incremental and primarily focused on application in the specific context of Sodankylä, Finland. Intuitively, an ensemble approach should outperform individual techniques; however, given the limited sample size—especially with SWE data (only around a dozen observations)—it becomes challenging to conclusively demonstrate superiority over simpler, more traditional methods such as multiple linear regression. Indeed, as highlighted in Table 3, some machine learning models significantly underperform in certain months, likely due to this limited dataset. Thus, at present, the main

takeaways and broader scientific significance are somewhat ambiguous. I encourage the authors to clearly articulate the core contributions of their approach, considering the constraints posed by dataset size. If a stronger case for novelty can be made, particularly in comparison to simpler or previously established methods, this would greatly strengthen the manuscript, as I am currently unsure of the main takeaways.

As noted in comment 3, we will add a convolutional neural network (CNN) model into the modeling approach. This deep learning model will serve as a contrast to the more commonly utilized and well-known machine learning methods and will serve to demonstrate how well each of these (RF, SVM, ANN, and MLR) perform against CNN. We will also be able to compare the CNN results to the machine learning-based EA results and identify which may be superior for snow depth or SWE estimation from our study site, with possible implications for other snow prone areas. The CNN model will also be included into the EA modeling in a hybrid-like approach of machine learning and deep learning, with it being possible to note how much of a positive influence, if any, this may have on the output metrics compared to a machine learning only EA model. A weighted, hybrid model that combines deep learning with traditional machine learning can attempt to fuse the intricacy of multiple layers of neural networks with the relative simplicity of more traditional models for snow depth estimation over a winter timeframe, before then estimating SWE with the predicted snow depth values. To aid in providing more valuable input data, we will generate polygons with a ~3 m radius at each observed field point that will contain average raster band values. Previously, the band values for each point were linked to image objects that could be long and thin, or have the point located near the corner or edge and thus not truly represent the band values of nearby features such as trees or water. This will ensure that the input data for this new modeling approach better incorporates the spatial context of surrounding features and will improve modeling performance.

**2.** Further clarity is needed regarding the training and validation processes for the machine learning models. The authors briefly mention using a "k-fold" validation but do not clearly specify how the data was partitioned into training, validation, and test sets at each step. Important details are missing, such as whether splits were random or sequential—random splits could inadvertently introduce spatial autocorrelation issues. Additionally, specifics on the machine learning implementations are essential. For instance, how deep were the random forest trees

allowed to grow? What structure was adopted for training the multi-layer perceptron—including the number of hidden layers, neurons per layer, activation functions, epochs, and optimization methods? Providing visualizations of training and validation curves for MLP models would also help clarify the model training and generalization processes. These details are crucial for reproducibility and fully understanding the robustness of the results.

Input data was randomly assigned into a 10-fold CV approach that separated the input data into a similar number of observations for the training or testing partitions. In each iteration, 90% of the data was utilized for training the model, while a different 10% was used to test that model. In the next iteration, a different 10% of the data would be used to test the model, with 90% of the data being for training. Thus, each observation is in the testing group 1 time and used to train the model 9 different times. Given that each iteration is run independently, each successive iteration does not result in the model learning from previous training/testing CV results which may result in biased outcomes. The outcomes are compared once all the iterations have been run with a mean of the model scores used for the final outcome. This will be included in the text. Chosen hyperparameter settings for each of the model inputs will be added into section 3.2 and explained how they contributed to model performance. In addition, the model parameters will also be placed into a table in the appendix to help with clarity about what was set for each model, as suggested by Reviewer #2.

**3.** Given the inherently spatial nature of snow depth and SWE, I'm curious if the authors considered employing machine learning methods specifically designed to leverage spatial dependencies in data. The current choice of models—MLR, RF, and MLP—generally treats each data point independently, potentially losing valuable spatial context unless explicitly provided as an input feature. Models that explicitly capture spatial information (e.g., convolutional neural networks like U-Nets, or vision transformer approaches) could better represent the spatial variability across diverse land types. Exploring spatially aware methods, despite your current dataset limitations, could significantly increase the novelty and impact of your study.

Currently our input data for snow depth and SWE contain raster values associated with image objects that may not properly account for the spatial variability of nearby features. For instance, the field data may have been in the corner of an image object, or in one that is long and thin, with either case not providing a true indication of the surrounding terrain in reference to the location

of the actual field data. To better integrate spatial context to the modeling procedure, we will instead generate a ~3 m radius polygon around each field measurement that will include the average and standard deviation of the raster data such as spaceborne imagery band values, elevation, and canopy height. This will allow nearby features which may have affected the real-world snow depth and SWE values to be better connected to the observed field snow depth and SWE data before then repeating the modeling. We will also include a CNN model into our approach to compare how this method compares with the previous models, how it can be integrated into the ensemble approach, and what benefit that may result for EA with estimating snow depth and SWE.

**4.** Finally, I also feel that this paper would really benefit from a more comprehensive comparison to existing approaches in the literature. Although your method is LiDAR-derived, related studies by Bair et al. (2018), King et al. (2020), Liljestrand et al. (2024), Shao et al. (2022), and Vafakhah et al. (2022) (amongst others) have utilized similar ML methodologies (RF and neural-network-based architectures) to predict regional variations of SWE. A clearer positioning of your work in relation to these papers would not only help justify the novelty of your method but also allow readers to better appreciate your contributions relative to the current state-of-the-art approaches. Such contextualization could also probably help address some of the concerns I raise in Comment 1 regarding methodological novelty.

The mentioned studies will be included in the updated manuscript, and it will be discussed how the methodology here compares with the methodologies presented. These studies and many like them (ours included) have noted the use of ML methodologies like RF, SVM, and other regression-based models in predicting snow related features, often with the support of various remote sensing data. Here we focused on comparing commonly utilized regression-based ML models and a weighted ensemble model to first estimate snow depth in six instances over a winter period, before then utilizing the more numerous snow depth data to aid in estimating more limited SWE data over the same period. To further distinguish our work, we will incorporate a deep learning CNN model for comparison to the ML models and integrate it into the weighted ensemble approach. Thus, the final model would be a weighted, hybrid ensemble approach of machine learning and deep learning.

**Minor Comments:**

- **Lines 89:** With all the different datasets being used here, I wonder if a summary table listing their names, variables, resolution, and source would help better situate readers?

  A summary table listing different data types, sources, names, resolutions, etc. will be provided in section 2.2 to help more clearly visualize the datasets used in this approach.

- **Lines 162-163:** It wasn't totally clear to me what this RF classification scheme was referring to here? Why is this step necessary?

  The acquired Land Use Land Cover (LULC) data, while very helpful, was limited at a 20 m resolution that was coarse for the chosen study site and limited the ability to make clear connections with obtained field data and vegetation types. In addition, the LULC data was from 2018, and may have become more outdated since that time. Thus, we needed to downscale the data to the 2 m resolution of the WorldView-2 imagery and the LiDAR data to provide a clear connection between these values and the landcover types. Just changing the resolution would result in many misclassifications, especially with artificial features and in heterogenous areas. As a result, we needed to utilize a classification-based scheme to better connect findings with properly downscaled and classified land cover types. While many different classification models would have served well, the best performance was obtained from Random Forest, which was why it was chosen for this purpose.

- **Section 3.1:** I also don't fully understand this image segmentation step and how it is "*utilized as the spatial unit for image assessment*". Why does this need to be done for this project, and how are the resulting segments used in the models afterwards?

  All pixels found in a specified image are separated into groupings of similar pixel values. These grouped pixels are then converted into polygons across the entire study area, with each polygon representing grouped pixels that match real-world features such as a cluster of bushes or a small body of water. Each of these polygons, now referred to as image objects, will then contain the average and standard deviation of all raster-based data inputs which are separated into columns. The field data are then placed into image objects based on spatial location and modeling is performed. This was accomplished for this

project as a pixel-based method can result in heightened variance in nearby pixel values that can result in extreme predictions, such as with the presence of shadows which were in the obtained imagery, or with rapid differences in predictions in very heterogenous areas. In addition, there may also be a potential mismatch between the imagery bands and the LiDAR data that may result in an individual pixel being incorrectly assigned a value. However, by using the image objects for the image assessment, it provides averaged band and LiDAR values that help to minimize extreme values, and thus provide more realistic values into the modeling approach. Once the modeling is completed, the predicted values are then assigned to all the image objects in the entire study area, as except for the field snow depth and SWE data, every image object contains the same set of raster data.

- **Lines 189-192:** I think this section is important, and I would add a little more detail describing each of these models and how they've been used in other studies, as they really underpin your main results. For instance, I'd mention bootstrapping and aggregation in the RF, and I would rework your description of the ANN (as the linkage to the human nervous system is somewhat spurious) and not a clear description of how it actually works (i.e., a feedforward directed acyclic graph connected with artificial neurons with nonlinear activation functions)

  We will expand upon the descriptions of each of the models listed in section 3.2 to provide greater clarity of what these models are and how they function. We will revise what was written for ANN and update it with text that better explains how it functions. Hyperparameters that were used in the models will also be included in this section.

- **Lines 203-204:** Do you know why the SVM performance so poor? I'm wondering if the sample was simply too small for this approach? This goes back to my earlier major point that the same issue with the limited SWE data is also likely impacting the other models. However, it does feel a bit odd to me to just choose to not include a model in some cases due to poor performance when using an ensemble approach

  SVM was dropped for SWE likely due to the lack of available field data, which as was seen with RF could result in poor performance, especially if outliers were present. As noted earlier, many of the field input data were joined to image objects but may have been spatially located in a corner or edge or be a part of a long and thin image object. All

of these may have not represented the true surroundings of the obtained field data. This will be addressed by changing the field inputs to collect raster values within a ~3 m radius of each field point, thereby providing proper spatial context into the modeling. It is also possible that hyperparameters were overlooked or not functioning as expected due to human error. Based on our findings, we will update our reasoning for the performance of SVM in the revised manuscript.

- **Eqs. 1/2/3:** This is personal preference but these are all very common metrics that don't need to be explicitly defined in this work

These three equations will be removed from the manuscript.

- **Lines 258-260:** From a physical perspective, what do you think is causing this large swing in performance for the ANN over these months? Is there something about the onset snow in December that makes this an especially challenging task for the NN?

A possibility is that in December, which is in the early middle of the winter period, there is relatively little snow. Snow depth is thus less variable and is somewhat more uniform across the landscape regardless of canopy cover or vegetation type when compared to further in the winter period such as in March or April. In addition, in late autumn and early winter temperatures may rise above freezing and rain events may also occur, both of which may reduce and flatten the snowpack. Over the course of the winter period, the effects of frequent snowfall and wind patterns may have led to more noticeable differences in snow depth based on the landcover.

- **Table 1:** For this table and the others after, I am wondering if this would be more interpretable as a bar graph? Comparing so many numbers in a table like this can bit a bit challenging

For the revised manuscript we will attempt to update these tables into graphs that convey the same information, but ideally in a manner that is more clearly legible. For Tables 1 and 3 these graphs will show the MAE and RMSE, alongside the coefficient of determination ($R^2$) instead of the Pearson's correlation ($r$) as requested by Reviewer #2. Tables 2 and 4 will likewise be updated into bar graphs that include the mean and

standard deviation values, and be color coded to match the LULC values in the respective maps for ease of comparison.

- **Table 2:** Similar to my previous table comment

See comment above for Table 1.

- **Figure 5:** The red->green color scheme for snow depth can be challenging to view for color blind individuals, and I would recommend moving to something more accessible

Thank you for pointing this out. The color scheme for snow depth will be updated from red->green to a different variation of blue->orange, as is seen in Figure 5 (h) to make it more accessible for color blind individuals. This same color scheme change will be applied to Figures 7 and 8 for consistency and ease of comparison between figures.

- **Lines 318-319:** Was the SVM left out because it had bad performance everywhere for SWE? As you state, the RF was also inconsistent for SWE prediction, but was still included in this part of the analysis

Correct, SVM was left out as it was largely producing poor metrics for SWE. It is valid that RF was also inconsistent with modeling results. It was chosen to remove SVM as neither individually nor when it was added into the ensemble analysis did it provide meaningful outcomes, and in all cases reduced ensemble performance. With RF, while it sometimes did result in poorer outcomes, it also had instances where it provided meaningful outcomes and benefited the ensemble analysis, and was thus included in the SWE modeling.

- **Lines 344-362:** I appreciate the detail the authors put into comparing SWE over various land cover types, however this section (and other similar paragraphs) are a bit challenging to parse in their current form. Currently, you list many statistics in a row, and it isn't fully clear to me what I am to take from all of these stats? I wonder if you could restructure these paragraphs to highlight the most important findings and relate those to what the predictive accuracy means for each land cover type?

We will revise how this paragraph and other paragraphs like it are structured to particularly highlight the most important findings, while minimizing or eliminating

findings that are minor or add little value. This will result in text that is more substantial with less repeated listings of statistics, and with a greater emphasis on connecting more notable findings to what the respective predictive accuracy indicates for the vegetative land cover types.

- **Lines 428-429:** When referring to EA here, it sounds as if it is it's own technique, but really it is just a combination of the MLR/RF/MLP. And this enhanced performance in the EA is because of high variability in individual models with biases which mostly cancel out resulting in a more stable prediction. So is this section speaking primarily to the high variability of individual models?

  Yes, the EA outputs are the result of a mixture of model outputs with high variability that can often cancel out, and thus generally lead to more stable predictions. The text in this section highlights that RF had the highest variation out of the listed models, specifically in terms of having a large range between positive and more discouraging modeling metrics, such as having a $r$ value of .05 and 0.71. None of the other models listed experienced such a dramatic variation, despite utilizing the same input data but in different instances. This contrasted with ANN, MLR, and EA which tended to be more stable in each instance. We will clarify this in the revised text.

- **Line 430:** I would reword this sentence "*EA consistently produced the best or second best metrics, and generally produced the best metrics*"

  This sentence will be reworded in the revised manuscript as "*EA consistently produced the best or second best metrics compared to the base models.*"

- **Lines 471-475:** Could you have included reanalysis estimates from say ERA5 to provide temperature, humidity and pressure data to your models? While coarse, this would perhaps give you some additional information about the surrounding environmental context at the time of observation?

  We will investigate providing data such as temperature, humidity, and pressure into our models at the times of observations whether from remote sensing or ground-based systems. Similar data was also available with instruments in and around the chosen study area, however there were very few, with some spaced far away from the field transects.

> Ideally, had such data been readily available at a finer resolution, we would have inputted it into our modeling as these factors can be related to the development and change of snow depth, SWE, and snow density over time. At minimum, including such data would allow for enhanced understanding of the surrounding environmental context at the time of field observations and would be worth including in the discussion.

- **Lines 501-502:** I would strongly recommend including some code for reproducing at least a subset of these results, perhaps in an interactive notebook uploaded to Google Colab with some test data? Then others could more easily test and build on what you have provided here

> In the revised manuscript, we will provide a link for code to be made available to the public. It will also include data so that others who are interested can test and build upon what was done in this manuscript, and to verify that the outcomes are reproducible.

**References**

Bair, E. H., Abreu Calfa, A., Rittger, K., & Dozier, J. (2018). Using machine learning for real-time estimates of snow water equivalent in the watersheds of Afghanistan. The Cryosphere, 12(5), 1579–1594. https://doi.org/10.5194/tc-12-1579-2018

King, F., Erler, A. R., Frey, S. K., & Fletcher, C. G. (2020). Application of machine learning techniques for regional bias correction of snow water equivalent estimates in Ontario, Canada. Hydrology and Earth System Sciences, 24(10), 4887–4902. https://doi.org/10.5194/hess-24-4887-2020

Liljestrand, D., Johnson, R., Skiles, S. M., Burian, S., & Christensen, J. (2024). Quantifying regional variability of machine-learning-based snow water equivalent estimates across the Western United States. Environmental Modelling & Software, 177, 106053. https://doi.org/10.1016/j.envsoft.2024.106053

Shao, D., Li, H., Wang, J., Hao, X., Che, T., & Ji, W. (2022). Reconstruction of a daily gridded snow water equivalent product for the land region above 45° N based on a ridge regression

machine learning approach. Earth System Science Data, 14(2), 795–809.
https://doi.org/10.5194/essd-14-795-2022

Vafakhah, M., Nasiri Khiavi, A., Janizadeh, S., & Ganjkhanlo, H. (2022). Evaluating different machine learning algorithms for snow water equivalent prediction. Earth Science Informatics, 15(4), 2431–2445. https://doi.org/10.1007/s12145-022-00846-z

Thank you for the references, these will be included and cited in the manuscript.

---

## Author Comment (AC2)

Dear Reviewer #2,

We greatly appreciate your time reviewing our manuscript and providing valuable feedback to aid in addressing weak points and strengthening the clarity and methodology of the manuscript. In the following, we address each of the points raised. Black text indicates the reviewer's comments. Blue text indicates our response and changes to the manuscript.

The paper "Object-based ensemble estimation of snow depth and snow water equivalent over multiple months in Sodankylä, Finland," authored by Brodylo et al., investigates the use of four machine learning techniques and their ensemble for snow depth estimation. The estimated snow depths were then used to estimate SWE. Finally, the ratio of the modeled SWE to snow depths was taken to estimate snow density. In my estimation, the paper is well written. However, I have major comments regarding the methodological clarity.

1. In section 3.2, the authors mentioned using Artificial Neural Networks (ANNs), among other models. However, they did not mention the exact architecture of the ANN (e.g., feed-forward, convolutional, transformers, etc.) used. Without this information, it is difficult to evaluate the appropriateness of the ANN architecture used in the study.

   The architecture of the ANN model was a feed-forward model with a single hidden layer. Specifically, we used "nnet" from the "caret" package in the R programming language. This will be expanded upon and added to the text in section 3.2.

2. In section 3.2, the details of the hyperparameters of the ML models (SVM, RF, and ANN) used were not mentioned. For example, for ANN, in addition to the architecture type, it would be beneficial to add the number of layers and neurons per layer, the activation function used, regularization (if any), the number of epochs, and other important hyperparameters used. For SVM, the kernel used, gamma, tolerance, and other important hyperparameters should be specified. For RF, the number of trees, the maximum depth, the minimum number of samples required to be at a leaf node, the minimum number of

samples required to split an internal node, and other important hyperparameters should be specified. These details are essential for reproducibility.

The hyperparameters for the RF, SVM, and ANN models will be added into section 3.2 to provide greater clarity for how the models were tuned and brief explanations for their purpose in improving model performance. This section will include the hyperparameters for the convolutional neural network (CNN) model as suggested by Reviewer #1. These will also be included in a table of chosen hyperparameters in the appendix for added legibility and clarity.

3. Also, in section 3.2, the authors mentioned using 10-fold cross-validation. However, important details are missing.

    1. Was the 10-fold CV done on the entire dataset or just the training set?

        The 10-fold CV was performed on the entire dataset to generate model performance when making predictions with the RF, SVM, ANN, and MLR models. For EA, it was indirectly applied, in that the outcomes were based off the weighted combination of base model outputs which were obtained with CV.

    2. No details about the train/test split ratio and strategy (random, stratified, etc) were mentioned.

        The utilized 10-fold CV approach was utilized to randomly separate the input data into either training or testing. In each iteration, 90% of the data was utilized for training the model, while a different 10% was used to test that model. In the next iteration, a different 10% of the data would be used to test the model, with 90% of the data being for training. Thus, each observation is in the testing group 1 time and used to train the model 9 different times. Given that each iteration is run independently, each successive iteration does not result in the model learning from previous training/testing CV results which may result in biased outcomes. The outcomes are compared once all the iterations have been run with a mean of the model scores used for the final outcome.

3. During the CV, how were hyperparameter configurations selected? Was it a grid search or Bayesian? A table of the hyperparameters tuned and their optimal values can be placed in the appendix.

   Hyperparameters were selected and configured based on a manual trial-and-error approach. We will include a table of chosen hyperparameters in the appendix of this work alongside their optimal values. This will be for all models including the addition of a convolutional neural network (CNN) model as suggested by Reviewer #1.

4. In section 3.3, the authors used Pearson's correlation as a measure of prediction accuracy. However, a perfect correlation does not necessarily mean that the model is good or that the predicted values are close to the true values. For example, $cor(y, y) = cor(y, 20y) = cor(y, 300y) = cor(y, 10000y) = 1$. That is to say, a model could be doing significantly worse and still have a perfect correlation. I encourage the authors to use the coefficient of determination instead. Please do not square the correlation coefficient; you can use r2_score in sklearn (https://scikit-learn.org/stable/modules/generated/sklearn.metrics.r2_score.html) or see this link for the formula (https://scikit-learn.org/stable/modules/model_evaluation.html#r2-score).

   For the measure of prediction accuracy, we will use the coefficient of determination ($r^2$) instead of the Pearson's correlation ($r$) for all models. As the code was produced in the R programming language, the coefficient of determination will be determined from "r.squared". This will ensure that the coefficient of determination is directly utilized and is not based upon squaring previously obtained correlation coefficient values.

5. This study uses 13 points of SWE and 88 points of depths to train the ML models. This is an extremely limited sample size for training any machine learning model, especially when trying to predict across 37,917 image objects with varying characteristics. This raises a serious concern about overfitting. With such a small training set, for example, for the SWE estimation problem, there's a high risk that the model would simply memorize the patterns in those 13 objects rather than learning generalizable relationships. Therefore, the authors should comment on how to validate the SWE across the upscaled

10 km$^2$. How did the authors ensure that the model wasn't overfitting for the SWE estimates? These points should be added to the discussion.

The issue of limited sample sizes, especially for SWE was considered during the development of this study. As the study site was in a 10 km$^2$ study area, this limited the influence of major weather patterns that would influence a portion of the study area, while not impacting other areas in the study. Here, if major weather events such as snowfall or change in temperature did occur, it would have been relatively uniform and over a similar start and stop period. In addition, land cover types were largely well covered with field snow depth data that may be related to SWE measurements, even if the SWE data was more limited. To address the overfitting issue, one method was using the cross-validation approach with 10 folds. With this, we could tune hyperparameters on the original training set, while also not letting the testing set be seen for selecting the final model. The cross-validation approach was also modified to minimized data leakage by incorporating preprocessing techniques such as standardization. In a simpler approach such as one time split between training and testing, the model could get lucky with a certain set of inputs for the testing/training phase, and then provide an unusually optimistic outcome that would give caution for overfitting or bias. If this occurs during the cross-validation approach that one iteration has extremely positive metrics, it would not be the final choice given that the final output would be an average of all 10 iterations, including those that provide poor metrics. In addition, multiple trial runs were done with a pseudorandom number generator when assigning the training / testing for the cross-validation modeling to verify if the outputted metrics were consistent or if there were any unusually high spikes. As multiple models were also tested in each instance, it was possible to compare the mapped results between all models. While in theory all models could be overfitted, they should ideally at minimum reveal similar trends in the data. If one or each model has notably differing results, it is a warning that something may not be right with the modeling and warrants further investigation. Lastly, given the limited data availability, model complexity was halted from become too elaborate and thus considering noise in the data that would invite the possibility of overfitting.

6. Line 204: The model weights should use another metric since correlation is not reliable based on comment 4. Also, I think adding the weighting formula would be helpful to readers.

   We will change the model weights to the coefficient of determination ($r^2$) based on comment 4. The weighting formula for how the ensemble analysis is performed will be placed in section 3.3 and explained.

7. Line 203: SVM was dropped due to poor performance. Could you please quantify "poor" in this scenario?

   Poor performance indicated instances where SVM provided unsatisfactory metric values that would have negatively impacted the ensemble analysis metrics and outcomes. SVM metric outputs for SWE estimation will be included and expanded upon in the revised manuscript.

8. Figure 3: One might think field snow depth and field swe are inputs. The authors should clarify in the caption that they are the outcome variables, not the input. Or they could represent output data with a different color.

   We will update Figure 3 to better reflect the distinction for snow depth and SWE being outcome variables and not as input variables. The figure caption mentions "Blue indicates input/output data", and we see how this can confuse readers or misrepresent the methodology framework given that this includes input and output data but does not clarify which is which in the figure. We will apply a unique color to field snow depth and SWE, and then clarify in the caption that these are output variables.

9. Tables 1-4: Were these metrics obtained from the entire dataset or just the testing set?

   The values in Tables 1 and 3 were obtained from the data inputs in the cross-validation approach. The field data values in Tables 2 and 4 were obtained from the entire data where field input values were available and matched to vegetative landcover types. Local scale outputs here were obtained from the entire dataset and matched to vegetative landcover types.

10. The authors should comment on the transferability of the ML models in this study. Can we grab this model and apply it elsewhere? The authors could dedicate a paragraph to model transferability in the discussion.

We will add a paragraph in the discussion that discusses the potential of the applied model to be utilized in other snow-prone regions of the world, and how this may be transferred to such regions. We will note the similarities in regions that have similar terrain, while also noting potential challenges in other snow-prone regions with differing conditions such as in mountainous terrain. In addition, it will be mentioned what data inputs would be necessary or beneficial to include to make the model function well.

11. Line 167: A period is missing between "scale" and "In OBIA".

Thank you, this will be corrected.

---

## Author Response (AR1)

Dear Editor and Reviewers,

We would like to first thank you for taking your time in reviewing our manuscript and providing constructive feedback to aid in addressing weak points and areas of concern, while also seeking to strengthen the novelty of the presented work. In the following, we addressed each of the points raised. Black text indicates the reviewer's comments. Blue text indicates our response and changes to the manuscript.

Your feedback is much appreciated.

Sincerely,
David Brodylo

**Reviewer #1**

**General Comments:**

Brodylo et al.'s manuscript is well-written, structured clearly, and supported by strong graphical presentation, providing a straightforward exploration into snow depth and snow water equivalent (SWE) estimation using an ensemble machine learning approach. The integration of LiDAR, remote sensing imagery, and in-situ observations is logical and aligns well with the type of studies frequently published in this journal. However, I have several significant concerns regarding the novelty of the approach, methodological clarity, and the limited sample size—particularly for SWE estimation—that need to be thoroughly addressed before the paper can be considered for publication. I have outlined these major concerns, along with specific suggestions for improvement, in detail below.

**Major Comments:**

1. Currently, the paper's primary novel contributions are unclear to me. While the presented approach effectively integrates established practices (ensemble machine learning methods, LiDAR-based snow depth estimation), the methodological novelty seems incremental and primarily focused on application in the specific context of Sodankylä, Finland. Intuitively, an ensemble approach should outperform individual techniques; however, given the limited sample size—especially with SWE data (only around a dozen observations)—it becomes challenging to conclusively demonstrate superiority over simpler, more traditional methods such as multiple linear regression. Indeed, as highlighted in Table 3, some machine learning models significantly underperform in certain months, likely due to this limited dataset. Thus, at present, the main takeaways and broader scientific significance are somewhat ambiguous. I encourage the authors to clearly articulate the core contributions of their approach, considering the constraints posed by

dataset size. If a stronger case for novelty can be made, particularly in comparison to simpler or previously established methods, this would greatly strengthen the manuscript, as I am currently unsure of the main takeaways.

We added a convolutional neural network (CNN) model into the modeling approach. This deep learning model served as a contrast to the more commonly utilized and well-known machine learning methods and served to demonstrate how well each of these (RF, SVM, ANN, and MLR) perform against CNN. CNN results were also compared to the machine learning-based EA results for snow depth or SWE estimation. The CNN model was also included into the EA modeling in a hybrid-like approach of machine learning and deep learning. A weighted, hybrid model that combines deep learning with traditional machine learning can attempt to fuse the intricacy of multiple layers of neural networks with the relative simplicity of more traditional models for snow depth estimation over a winter timeframe, before then estimating SWE with the predicted snow depth values. To aid in providing more valuable input data, we generated polygons with a 3 m radius at each observed field point that contain average and standard deviation raster band values. Previously, the band values for each point were linked to image objects that could be long and thin, or have the point located near the corner or edge and thus not truly represent the band values of nearby features such as trees or water. This ensured that the input data for this new modeling approach better incorporated the spatial context of surrounding features and led to notable modeling performance, especially for SVM and RF.

2. Further clarity is needed regarding the training and validation processes for the machine learning models. The authors briefly mention using a "k-fold" validation but do not clearly specify how the data was partitioned into training, validation, and test sets at each step. Important details are missing, such as whether splits were random or sequential—random splits could inadvertently introduce spatial autocorrelation issues. Additionally, specifics on the machine learning implementations are essential. For instance, how deep were the random forest trees allowed to grow? What structure was adopted for training the multi-layer perceptron—including the number of hidden layers, neurons per layer, activation functions, epochs, and optimization methods? Providing visualizations of training and validation curves for MLP models would also help clarify the model training and generalization processes. These details are crucial for reproducibility and fully understanding the robustness of the results.

Input data was randomly assigned into a 10-fold CV approach that separated the input data into a similar number of observations for the training or testing partitions. In each iteration, 90% of the data was utilized for training the model, while a different 10% was used to test that model. In the next iteration, a different 10% of the data would be used to test the model, with 90% of the data being for training. Thus, each observation is in the testing group 1 time and used to train the model 9 different times. Given that each iteration is run independently, each successive iteration does not result in the model learning from previous training/testing CV results which may result in biased outcomes. The outcomes are compared once all the iterations have been run with a mean of the model scores used for the final outcome. Chosen hyperparameter settings for each of the model inputs were added into the appendix as suggested by Reviewer #2.

**3.** Given the inherently spatial nature of snow depth and SWE, I'm curious if the authors considered employing machine learning methods specifically designed to leverage spatial dependencies in data. The current choice of models—MLR, RF, and MLP—generally treats each data point independently, potentially losing valuable spatial context unless explicitly provided as

an input feature. Models that explicitly capture spatial information (e.g., convolutional neural networks like U-Nets, or vision transformer approaches) could better represent the spatial variability across diverse land types. Exploring spatially aware methods, despite your current dataset limitations, could significantly increase the novelty and impact of your study.

Previously, our input data for snow depth and SWE contained raster values associated with image objects that may not properly account for the spatial variability of nearby features. For instance, the field data may have been in the corner of an image object, or in one that is long and thin, with either case not providing a true indication of the surrounding terrain in reference to the location of the actual field data. To better integrate spatial context to the modeling procedure, we instead generated 3 m radius polygons around each field measurement that included the average and standard deviation of the raster data such as spaceborne imagery band values, elevation, and canopy height. This allowed nearby features which may have affected the real-world snow depth and SWE values to be better connected to the observed field snow depth and SWE data before then repeating the modeling. We also include a CNN model into our approach to compare how this method compares with the previous models, how it can be integrated into the ensemble approach with a hybrid of machine and deep learning, and what benefit may result for EA with estimating snow depth and SWE.

**4.** Finally, I also feel that this paper would really benefit from a more comprehensive comparison to existing approaches in the literature. Although your method is LiDAR-derived, related studies by Bair et al. (2018), King et al. (2020), Liljestrand et al. (2024), Shao et al. (2022), and Vafakhah et al. (2022) (amongst others) have utilized similar ML methodologies (RF and neuralnetwork-based architectures) to predict regional variations of SWE. A clearer positioning of your work in relation to these papers would not only help justify the novelty of your method but also allow readers to better appreciate your contributions relative to the current state-of-the-art approaches. Such contextualization could also probably help address some of the concerns I raise in Comment 1 regarding methodological novelty.

The mentioned studies were included in the updated manuscript. These studies and many like them (ours included) have noted the use of ML methodologies like RF, SVM, and other regression-based models in predicting snow related features, often with the support of various remote sensing data. Here we focused on comparing commonly utilized regression-based ML models and a weighted ensemble model to first estimate snow depth in six instances over a winter period, before then utilizing the more numerous snow depth data to aid in estimating more limited SWE data over the same period. To further distinguish our work, we incorporated a deep learning CNN model for comparison to the ML models and integrated it into the weighted ensemble approach. Thus, the final model would be a weighted, hybrid ensemble approach of machine learning and deep learning.

**Minor Comments:**

• Lines 89: With all the different datasets being used here, I wonder if a summary table listing their names, variables, resolution, and source would help better situate readers?

A summary table listing different data types, sources, names, resolutions, etc. was provided at the end of section 2.2 to help more clearly visualize the datasets used in this approach.

• Lines 162-163: It wasn't totally clear to me what this RF classification scheme was referring to here? Why is this step necessary?

The acquired Land Use Land Cover (LULC) data, while very helpful, was limited at a 20 m resolution that was coarse for the chosen study site and limited the ability to make clear connections with obtained field data and vegetation types. In addition, the LULC data was from 2018, and may have become more outdated since that time. Thus, we needed to downscale the data to the 2 m resolution of the WorldView-2 imagery and the LiDAR data to provide a clear connection between these values and the landcover types. Just changing the resolution would result in many misclassifications, especially with artificial features and in heterogenous areas. As a result, we needed to utilize a classification-based scheme to better connect findings with properly downscaled and classified land cover types. While many different classification models would have served well, the best performance was obtained from Random Forest, which was why it was chosen for this purpose.

• Section 3.1: I also don't fully understand this image segmentation step and how it is "utilized as the spatial unit for image assessment". Why does this need to be done for this project, and how are the resulting segments used in the models afterwards?

All pixels found in a specified image are separated into groupings of similar pixel values. These grouped pixels are then converted into polygons across the entire study area, with each polygon representing grouped pixels that match real-world features such as a cluster of bushes or a small body of water. Each of these polygons, now referred to as image objects, will then contain the average and standard deviation of all raster-based data inputs which are separated into columns. The field data are then placed into image objects based on spatial location and modeling is performed. This was accomplished for this project as a pixel-based method can result in heightened variance in nearby pixel values that can result in extreme predictions, such as with the presence of shadows which were in the obtained imagery, or with rapid differences in predictions in very heterogenous areas. In addition, there may also be a potential mismatch between the imagery bands and the LiDAR data that may result in an individual pixel being incorrectly assigned a value. However, by using the image objects for the image assessment, it provides averaged band and LiDAR values that help to minimize extreme values, and thus provide more realistic values into the modeling approach. Once the modeling is completed, the predicted values are then assigned to all the image objects in the entire study area, as except for the field snow depth and SWE data, every image object contains the same set of raster data.

• Lines 189-192: I think this section is important, and I would add a little more detail describing each of these models and how they've been used in other studies, as they really underpin your main results. For instance, I'd mention bootstrapping and aggregation in the RF, and I would rework your description of the ANN (as the linkage to the human nervous system is somewhat spurious) and not a clear description of how it actually works (i.e., a feedforward directed acyclic graph connected with artificial neurons with nonlinear activation functions)

We expanded upon the descriptions of each of the models listed in section 3.2 and in the appendix to provide greater clarity of what these models are and how they function. We adjusted the description for ANN and updated it with text that better explains how it functions.

Lines 203-204: Do you know why the SVM performance so poor? I'm wondering if the sample was simply too small for this approach? This goes back to my earlier major point that the same issue with the limited SWE data is also likely impacting the other models. However, it does feel a bit odd to me to just choose to not include a model in some cases due to poor performance when using an ensemble approach

SVM was dropped for SWE at the time likely due to the lack of available field data, which as was seen with RF could result in poor performance, especially if outliers were present. As noted earlier, many of the field input data were joined to image objects but may have been spatially located in a corner or edge or be a part of a long and thin image object. All of these may have not represented the true surroundings of the obtained field data. This was addressed by changing the field inputs to collect raster values within a 3 m radius of each field point, thereby providing proper spatial context into the modeling. This helped lead to a substantial improvement in SVM performance and including it in the ensemble model, along with for the other models. Hyperparameters were further tweaked because of this change.

• Eqs. 1/2/3: This is personal preference but these are all very common metrics that don't need to be explicitly defined in this work

These three equations were removed from the manuscript.

• Lines 258-260: From a physical perspective, what do you think is causing this large swing in performance for the ANN over these months? Is there something about the onset snow in December that makes this an especially challenging task for the NN?

A possibility is that in December, which is in the early middle of the winter period, there is relatively little snow. Snow depth is thus less variable and is somewhat more uniform across the landscape regardless of canopy cover or vegetation type when compared to further in the winter period such as in March or April. In addition, in late autumn and early winter temperatures may rise above freezing and rain events may also occur, both of which may reduce and flatten the snowpack. Over the course of the winter period, the effects of frequent snowfall and wind patterns may have led to more noticeable differences in snow depth based on the landcover.

• **Table 1:** For this table and the others after, I am wondering if this would be more interpretable as a bar graph? Comparing so many numbers in a table like this can bit a bit challenging

For the revised manuscript we updated these tables into graphs that convey the same information, but ideally in a manner that is more clearly legible. For Tables 1 and 3 these graphs show the MAE and RMSE, alongside the coefficient of determination ( $R^2$ ) instead of the Pearson's correlation (r) as requested by Reviewer #2. Tables 2, 4, and 6 were likewise updated into bar graphs that include the mean and standard deviation values, and were color coded to match the LULC values in the respective maps for ease of

comparison. Maps showing the snow depth, SWE, and snow density were all also updated with the updated EA based results.

• Table 2: Similar to my previous table comment

See comment above for Table 1.

• **Figure 5:** The red->green color scheme for snow depth can be challenging to view for color blind individuals, and I would recommend moving to something more accessible

Thank you for pointing this out. The color scheme for snow depth was now updated from red -> green to a different variation of blue->orange, as is seen in the previous version of Figure 5 (h) to make it more accessible for color blind individuals. This same color scheme change was applied to Figures 7 and 8 for consistency and ease of comparison between figures.

• Lines 318-319: Was the SVM left out because it had bad performance everywhere for SWE? As you state, the RF was also inconsistent for SWE prediction, but was still included in this part of the analysis

Correct, SVM was left out as it was largely producing poor metrics for SWE. It is valid that RF was also inconsistent with modeling results. It was chosen to remove SVM at the time as neither individually nor when it was added into the ensemble analysis did it provide meaningful outcomes, and in all cases reduced ensemble performance. With RF, while it sometimes did result in poorer outcomes, it also had instances where it provided meaningful outcomes and benefited the ensemble analysis, and was thus included in the SWE modeling. However, due to undergoing notable field data image object changes, we have redone model performance and have included SVM for SWE. This is largely due to changing the field-based image objects to instead be polygons with a 3 m radius at each field location, which thereby better included nearby terrain and vegetation values. While this benefited all models, it was most evident with SVM.

• Lines 344-362: I appreciate the detail the authors put into comparing SWE over various land cover types, however this section (and other similar paragraphs) are a bit challenging to parse in their current form. Currently, you list many statistics in a row, and it isn't fully clear to me what I am to take from all of these stats? I wonder if you could restructure these paragraphs to highlight the most important findings and relate those to what the predictive accuracy means for each land cover type?

We revised how this paragraph and other paragraphs like it were structured to highlight the most important findings, while minimizing or eliminating findings that are minor or add little value. The repetition of values was reduced to make it clearer. This was done through most of the results section of the manuscript.

• Lines 428-429: When referring to EA here, it sounds as if it is it's own technique, but really it is just a combination of the MLR/RF/MLP. And this enhanced performance in the EA is because of high variability in individual models with biases which mostly cancel out resulting in a more stable prediction. So is this section speaking primarily to the high variability of individual models?

Yes, the EA outputs are the result of a mixture of model outputs with high variability that can often cancel out, and thus generally lead to more stable predictions. The text in this section highlights that RF had the highest variation out of the listed models, specifically in terms of having a large range between positive and more discouraging modeling metrics, such as having a r value of .05 and 0.71. None of the other models listed experienced such a dramatic variation, despite utilizing the same input data but in different instances. This contrasted with ANN, MLR, and EA which tended to be more stable in each instance. In the current manuscript this portion was updated with the inclusion of the CNN and SVM models, alongside updated metrics for all models.

• **Line 430:** I would reword this sentence "EA consistently produced the best or second best metrics, and generally produced the best metrics"

This sentence was reworded in the revised manuscript as "EA contained the most stable and positive metrics for  $R^2$  in all instances."

• Lines 471-475: Could you have included reanalysis estimates from say ERA5 to provide temperature, humidity and pressure data to your models? While coarse, this would perhaps give you some additional information about the surrounding environmental context at the time of observation?

We attempted to include data from ERA5, and while it did provide some additional context, it was still nonetheless hard to visually distinguish how factors such as temperature, humidity and pressure with a coarser resolution could be immediately made relevant. When inputted into the models, at the best it was mixed results while oftentimes limiting performance. Had our field locations been more dispersed and greater in number, it likely would have aided the modeling performance.

• Lines 501-502: I would strongly recommend including some code for reproducing at least a subset of these results, perhaps in an interactive notebook uploaded to Google Colab with some test data? Then others could more easily test and build on what you have provided here

We provided a link for code to be made available to the public near the bottom of the manuscript in the section "Code and data availability". It also includes data so that others who are interested can test and build upon what was done in this manuscript, and to verify that the outcomes are reproducible.

**References**

Bair, E. H., Abreu Calfa, A., Rittger, K., & Dozier, J. (2018). Using machine learning for real-time estimates of snow water equivalent in the watersheds of Afghanistan. The Cryosphere, 12(5), 1579–1594. <a href="https://doi.org/10.5194/tc-12-1579-2018">https://doi.org/10.5194/tc-12-1579-2018</a>

King, F., Erler, A. R., Frey, S. K., & Fletcher, C. G. (2020). Application of machine learning techniques for regional bias correction of snow water equivalent estimates in Ontario, Canada. Hydrology and Earth System Sciences, 24(10), 4887–4902. <a href="https://doi.org/10.5194/hess-24-4887-2020">https://doi.org/10.5194/hess-24-4887-2020</a>

Liljestrand, D., Johnson, R., Skiles, S. M., Burian, S., & Christensen, J. (2024). Quantifying regional variability of machine-learning-based snow water equivalent estimates across the

Western United States. Environmental Modelling & Software, 177, 106053. https://doi.org/10.1016/j.envsoft.2024.106053

Shao, D., Li, H., Wang, J., Hao, X., Che, T., & Ji, W. (2022). Reconstruction of a daily gridded snow water equivalent product for the land region above 45° N based on a ridge regression machine learning approach. Earth System Science Data, 14(2), 795–809. <a href="https://doi.org/10.5194/essd-14-795-2022">https://doi.org/10.5194/essd-14-795-2022</a>

Vafakhah, M., Nasiri Khiavi, A., Janizadeh, S., & Ganjkhanlo, H. (2022). Evaluating different machine learning algorithms for snow water equivalent prediction. Earth Science Informatics, 15(4), 2431–2445. https://doi.org/10.1007/s12145-022-00846-z

Thank you for the references, these were included and cited in the manuscript.

**Reviewer #2**

The paper "Object-based ensemble estimation of snow depth and snow water equivalent over multiple months in Sodankylä, Finland," authored by Brodylo et al., investigates the use of four machine learning techniques and their ensemble for snow depth estimation. The estimated snow depths were then used to estimate SWE. Finally, the ratio of the modeled SWE to snow depths was taken to estimate snow density. In my estimation, the paper is well written. However, I have major comments regarding the methodological clarity.

- 1. In section 3.2, the authors mentioned using Artificial Neural Networks (ANNs), among other models. However, they did not mention the exact architecture of the ANN (e.g., feed-forward, convolutional, transformers, etc.) used. Without this information, it is difficult to evaluate the appropriateness of the ANN architecture used in the study.
  - The architecture of the ANN model was a feed-forward model with a single hidden layer. Specifically, we used "nnet" from the "caret" package in the R programming language. This was expanded upon and added to the text in section 3.2 alongside in the appendix.
- 2. In section 3.2, the details of the hyperparameters of the ML models (SVM, RF, and ANN) used were not mentioned. For example, for ANN, in addition to the architecture type, it would be beneficial to add the number of layers and neurons per layer, the activation function used, regularization (if any), the number of epochs, and other important hyperparameters used. For SVM, the kernel used, gamma, tolerance, and other important hyperparameters should be specified. For RF, the number of trees, the maximum depth, the minimum number of samples required to be at a leaf node, the minimum number of samples required to split an internal node, and other important hyperparameters should be specified. These details are essential for reproducibility.

The hyperparameters for the RF, SVM, and ANN models were added into the appendix at the end to provide greater clarity for how the models were tuned and what values were specified. This section also includes the hyperparameters for the convolutional neural network (CNN) model as suggested by Reviewer #1.

- 3. Also, in section 3.2, the authors mentioned using 10-fold cross-validation. However, important details are missing.
  - 1. Was the 10-fold CV done on the entire dataset or just the training set?

The 10-fold CV was performed on the entire dataset to generate model performance when making predictions with the CNN, RF, SVM, ANN, and MLR models. For EA, it was indirectly applied, in that the outcomes were based off the weighted combination of base model outputs which were obtained with CV.

2. No details about the train/test split ratio and strategy (random, stratified, etc) were mentioned.

The utilized 10-fold CV approach was utilized to randomly separate the input data into either training or testing. In each iteration, 90% of the data was utilized for training the model, while a different 10% was used to test that model. In the next iteration, a different 10% of the data would be used to test the model, with 90% of the data being for training. Thus, each observation is in the testing group 1 time and used to train the model 9 different times. Given that each iteration is run independently, each successive iteration does not result in the model learning from previous training/testing CV results which may result in biased outcomes. The outcomes are compared once all the iterations have been run with a mean of the model scores used for the final outcome.

3. During the CV, how were hyperparameter configurations selected? Was it a grid search or Bayesian? A table of the hyperparameters tuned and their optimal values can be placed in the appendix.

Hyperparameters were selected and configured based on a manual trial-and-error approach. We included a table of chosen hyperparameters in the appendix of this work alongside utilized values. This is for all models including the addition of a convolutional neural network (CNN) model as suggested by Reviewer #1.

4. In section 3.3, the authors used Pearson's correlation as a measure of prediction accuracy. However, a perfect correlation does not necessarily mean that the model is good or that the predicted values are close to the true values. For example, cor(y, y) = cor(y, 20y) = cor(y, 300y) = cor(y, 10000y) = 1. That is to say, a model could be doing significantly worse and still have a perfect correlation. I encourage the authors to use the coefficient of determination instead. Please do not square the correlation coefficient; you can use r2\_score in sklearn (https://scikitlearn.org/stable/modules/generated/sklearn.metrics.r2\_score.html) or see this link for the formula (https://scikit-learn.org/stable/modules/model\_evaluation.html#r2-score).

For the measure of prediction accuracy, we used the coefficient of determination (R2) instead of the Pearson's correlation (r) for all models. As the code was produced in the R programming language, the coefficient of determination was determined from "r.squared". This ensures that the coefficient of determination is directly utilized and is not based upon squaring previously obtained correlation coefficient values.

5. This study uses 13 points of SWE and 88 points of depths to train the ML models. This is an extremely limited sample size for training any machine learning model, especially when trying to predict across 37,917 image objects with varying characteristics. This raises a serious concern about overfitting. With such a small training set, for example, for the SWE estimation problem, there's a high risk that the model would simply memorize the patterns in those 13 objects rather than learning generalizable relationships. Therefore, the authors should comment on how to validate the SWE across the upscaled 10 km². How did the authors ensure that the model wasn't overfitting for the SWE estimates? These points should be added to the discussion.

The issue of limited sample sizes, especially for SWE was considered during the development of this study. As the study site was in a 10 km2 study area, this limited the influence of major weather patterns that would influence a portion of the study area, while not impacting other areas in the study. Here, if major weather events such as snowfall or change in temperature did occur, it would have been relatively uniform and over a similar start and stop period. In addition, land cover types were largely well covered with field snow depth data that may be related to SWE measurements, even if the SWE data was more limited. To address the overfitting issue, one method was using the cross-validation approach with 10 folds. With this, we could tune hyperparameters on the original training set, while also not letting the testing set be seen for selecting the final model. In a simpler approach such as one time split between training and testing, the model could get lucky with a certain set of inputs for the testing/training phase, and then provide an unusually optimistic outcome that would give caution for overfitting or bias. If this occurs during the cross-validation approach that one iteration has extremely positive metrics, it would not be the final choice given that the final output would be an average of all 10 iterations, including those that provide poor metrics. In addition, multiple trial runs were done with a pseudorandom number generator when assigning the training / testing for the cross-validation modeling to verify if the outputted metrics were consistent or if there were any unusually high spikes. As multiple models were also tested in each instance, it was possible to compare the mapped results between all models. While in theory all models could be overfitted, they should ideally at minimum reveal similar trends in the data. If one or each model has notably differing results, it is a warning that something may not be right with the modeling and warrants further investigation. Given the limited data availability, model complexity was halted from become too elaborate and thus considering noise in the data that would invite the possibility of overfitting. Input data was also standardized and Principal Component Analysis (PCA) was performed, which aided to lessen overfitting and minimized data leakage.

- 6. Line 204: The model weights should use another metric since correlation is not reliable based on comment 4. Also, I think adding the weighting formula would be helpful to readers.
  - We changed the model weights to the coefficient of determination (R2) based on comment 4. The weighting formula for how the ensemble analysis is placed in section 3.3 and explained.
- 7. Line 203: SVM was dropped due to poor performance. Could you please quantify "poor" in this scenario?

Due to undergoing notable field data image object changes, we have redone model performance and have included SVM for SWE. This is largely due to changing the field-based image objects to instead be polygons with a 3 m radius at each field location, which thereby better included nearby terrain and vegetation values. Because of the change, hyperparameters were also tweaked for SVM, along with for ANN and RF.

8. Figure 3: One might think field snow depth and field swe are inputs. The authors should clarify in the caption that they are the outcome variables, not the input. Or they could represent output data with a different color.

Figure 3 was updated to better reflect the distinction for snow depth and SWE being outcome variables and not as input variables. A unique color for snow depth and SWE was applied, and it was clarified in the caption that these are output variables to be predicted and not the input.

9. Tables 1-4: Were these metrics obtained from the entire dataset or just the testing set?

The values in Tables 1 and 3 were obtained from the data inputs in the cross-validation approach. The field data values in Tables 2 and 4 were obtained from the entire data where field input values were available and matched to vegetative landcover types. Local scale outputs here were obtained from the entire dataset and matched to vegetative landcover types.

10. The authors should comment on the transferability of the ML models in this study. Can we grab this model and apply it elsewhere? The authors could dedicate a paragraph to model transferability in the discussion.

A paragraph in the discussion was added that discusses the potential of the applied model to be utilized in other snow-prone regions of the world, and how this may be transferred to such regions. We noted the similarities in regions that have similar terrain, while also noting potential challenges in other snow-prone regions with differing conditions such as in mountainous terrain. In addition, it was noted how the model had potential to adapt when sample sizes are low (simpler models) or when sample sizes are high (complex models).

11. Line 167: A period is missing between "scale" and "In OBIA".

Thank you, this has been corrected.

---

## Referee Report (RR1)

The authors have greatly improved the quality of the manuscript. Here are some comments to enhance the scientific rigor:

**Data Preprocessing (L215-216):**

- **Standardization**: The authors mentioned that all inputs were standardized, but did not specify the standardization method used. Was it Z-score, min-max scaling, or robust scaling?
- PCA: The authors mentioned that PCA was used, but provided no implementation details. How many principal components (PCs) were used for the machine learning model fitting? Were the PCs selected based on explained variance? If yes, what threshold was used? The PCA step should also be represented in the flowchart in Figure 3, as the authors' statement about using PCA (L215-216) suggests that the raw data weren't used as features in the ML models but PCs. This information is important because it helps the reader assess the severity of overfitting due to the limited SWE samples.

**Neural Network Architectures (Section 3.2):**

- **ANN vs FFN**: The authors clearly specified that the ANN model used in the manuscript was a feed-forward model (L205). However, because all deep learning models are considered ANNs, I suggest that the authors change ANN to feed-forward network (FFN).
- Lack of details about CNN: The only description of CNN given by the authors was on L207-208. While the description is technically accurate, it leaves a crucial detail about the dimension of the CNN used. Was it 1D, 2D, or 3D CNN. The parameters in Table 1 of Appendix A suggest that a 1D CNN was used. However, these details could be mentioned in section 3.2, as not all readers read appendices. Most readers would assume a 2D CNN for image analysis, making the 1D architecture choice counterintuitive without explanation. Additionally, a brief justification would strengthen the rationale for a 1D CNN.

**Metrics:**

- **R**2 **vs. r**2: It appears that the coefficient of determination reported in this study was obtained from regressing the observed SWE/depth on the predicted depth. This was confirmed from Line 114 of Snow\_Depth\_Code.R of the source code. Mathematically, this is equivalent to the square of a Pearson correlation coefficient (**r**2) between the observed and the predicted values. With this approach, there is a potential for overestimating the model performance. The reason is that as long as the observed and the predicted values trend together, the **r**2 will be high. For example, cor(y, y)^2 = cor(y, 20y)^2 = cor(y, 300y)^2 = cor(y, 10000y)^2 = 1. That is to say, a model could be doing significantly worse and still have a perfect **r**2. What should be reported is R2: 1 SSR/SST. The question R2 answers is: "If I compare these predictions directly to the true values, how close are they to lying on the perfect 1:1 line?" That would give us a more accurate assessment of predictive accuracy.
- **K-fold metric vs. overall metric**: According to L213 L214, metrics are determined from the average of the test folds. However, Figures 5 and 9 suggest that the coefficient of determination was obtained after the final predictions. Can the authors clarify this?

**L449**: The authors mentioned that "In comparison to the snow depth, there was a much smaller sample size, which led to greater model uncertainty and disagreement." I am not clear what disagreement is referring to in this situation. Is it disagreement between the different models or disagreement between the observed and the predicted SWE values? If the latter, the disagreement aspect doesn't seem supported by the results in the study. According to Figure 5, the coefficient of determination ranges between 0.86 and 0.92 for snow depth, but 0.93 and 0.97 (Figure 9).

**L508**: Could the authors clarify what kind of models they refer to as "uncomplicated?" Examples would be helpful.

**Table vs Figures**: Figures 4 and 8 could benefit from a tabular presentation.

**Table 1 in Appendix A:** the parameter column displays inconsistent information. For CNN, the column shows the actual layer configuration used, while for RF, SVM, and ANN, it displays the hyperparameter search grid rather than the selected values. The authors should standardize this

column to either show: (1) the final selected hyperparameters for all models, or (2) create separate columns for "hyperparameter search space" and "selected values."